# Comprehensive Review: High-Performance Positioning Systems for Navigation and Wayfinding for Visually Impaired People

**DOI:** 10.3390/s24217020

**Published:** 2024-10-31

**Authors:** Jean Marc Feghali, Cheng Feng, Arnab Majumdar, Washington Yotto Ochieng

**Affiliations:** 1Department of Civil and Environmental Engineering, Imperial College London, Skempton Building, South Kensington, London SW7 2BU, UK; jean.feghali14@imperial.ac.uk (J.M.F.); w.ochieng@imperial.ac.uk (W.Y.O.); 2Research and Development, WeWALK Smart Cane, London SW1W 9SH, UK

**Keywords:** visually impaired, mobility, positioning, navigation, wayfinding, PNT

## Abstract

The global increase in the population of Visually Impaired People (VIPs) underscores the rapidly growing demand for a robust navigation system to provide safe navigation in diverse environments. State-of-the-art VIP navigation systems cannot achieve the required performance (accuracy, integrity, availability, and integrity) because of insufficient positioning capabilities and unreliable investigations of transition areas and complex environments (indoor, outdoor, and urban). The primary reason for these challenges lies in the segregation of Visual Impairment (VI) research within medical and engineering disciplines, impeding technology developers’ access to comprehensive user requirements. To bridge this gap, this paper conducts a comprehensive review covering global classifications of VI, international and regional standards for VIP navigation, fundamental VIP requirements, experimentation on VIP behavior, an evaluation of state-of-the-art positioning systems for VIP navigation and wayfinding, and ways to overcome difficulties during exceptional times such as COVID-19. This review identifies current research gaps, offering insights into areas requiring advancements. Future work and recommendations are presented to enhance VIP mobility, enable daily activities, and promote societal integration. This paper addresses the urgent need for high-performance navigation systems for the growing population of VIPs, highlighting the limitations of current technologies in complex environments. Through a comprehensive review of VI classifications, VIPs’ navigation standards, user requirements, and positioning systems, this paper identifies research gaps and offers recommendations to improve VIP mobility and societal integration.

## 1. Introduction

Approximately 2.2 billion people around the world have near or distant vision impairment [1]. Visual Impairment (VI) reduces quality of life and lowers workforce participation and productivity. The World Health Organization estimates an annual global productivity loss of US$411 billion due to VI [1].

Currently, the most widely used aids for Visually Impaired People (VIPs) are white canes and guide dogs. However, these two aids are unable to provide safe and accurate navigation as neither can identify complex street signs or scenarios. Consequently, a significant challenge remains for VIPs to navigate independently. Considerable efforts are therefore being made to find a solution to VIPs’ navigation and wayfinding, initially focusing on Positioning, Navigation, and Timing (PNT) sensors.

Such efforts in seeking to develop high-performance mobility aids, however, suffer from the existing limitations of sensors. For example, while the main PNT sensor, Global Navigation Satellite System (GNSS), can deliver PNT information for users worldwide, with meter positioning accuracy using pseudo-range measurements and centimeter positioning accuracy using carrier phase measurements in real-time, it is unavailable both indoors and in urban environments where GNSS signals can be reflected, obstructed, and attenuated. This leads to potential errors and outages in the provision of PNT information for navigation. Alternatives to GNSS, such as dead reckoning, indoor positioning, and computer vision, can address these limitations. However, challenges in their implementation encompass issues such as inaccurate stereo cameras, reduced visual sensor performance in low-light conditions, and the significant size of Lidar sensors, among others.

An initial hurdle that must be surmounted in developing a high-performance positioning system for navigation and wayfinding for VIPs is the segregation of visually impaired research between medical and engineering disciplines. As an initial suggestion, promoting interdisciplinary collaboration among experts from diverse fields can enhance navigation solutions for VIPs.

To bridge this gap comprehensively, Section 2 of this paper delineates the global definitions of VI to provide a greater understanding of its spectrum. Section 3 outlines the principal international and regional standards of navigation for VIPs. Section 4 explores VIP mobility requirements across environmental, system, and user aspects. A notable research gap emerges concerning the understanding of VIP mobility requirements, particularly the lack of behavioral data in individual and crowd scenarios. To address this deficiency, a series of experiments are conducted at the Pedestrian Accessibility Movement Environment Laboratory (PAMELA), and the findings, which can serve as a foundation for future research endeavors, are highlighted in Section 5. These insights play a pivotal role in enhancing and extending environmental features and indoor navigation systems.

Section 6 scrutinizes existing positioning and navigation systems designed for VIPs and identifies seven limitations in current navigation aids for VIPs, encompassing positioning accuracy, gaps in transition areas, user comfort, motivation, multi-mode feedback, complex object detection, and social concerns such as scalability, cost-effectiveness, and privacy leakage. Notably, a quantitative analysis of these limitations is lacking, underscoring a critical research gap. Section 7 discusses the impact of pandemics, such as COVID-19, on the daily lives of VIPs. It highlights the relevant technologies developed to help overcome the challenges associated with navigation during such times.

The inclusion criteria include literature studies related to visual impairment classification, navigation systems, mobility aids, AI assistants, and the impact of COVID-19 on VIPs. Studies involving outdated technologies or lacking validation are excluded. A systematic search, employing terms such as “visually impaired” and “wayfinding,” spans academic and public databases, with efforts to minimize bias through multiple reviewers and thorough exploration of the literature. Figure 1 presents the methodological framework for the comprehensive literature review. The key methods of this review are outlined below:Review Justification: Section 1 delves into the navigation challenges faced by Visually Impaired People (VIPs);Precise Review Objective: This aims to develop a high-performance positioning system for VIPs, addressing the segregation of visually impaired research between medical and engineering disciplines;Inclusion and Exclusion Criteria: The included literature covers topics such as ‘classification of visual impairment’, ‘standards of navigation systems for visually impaired people’, ‘navigation assistants for visually impaired people’, ‘mobility aids for visually impaired people’, ‘positioning and wayfinding of navigation systems’, ‘artificial intelligence assistants for visually impaired people’, and ‘impact of COVID-19 on visually impaired people’. The excluded literature consists of outdated technological interventions and studies lacking validation;Explicit Literature Search: Key search terms include ‘visually impaired’, ‘mobility aids’, ‘navigation’, ‘positioning’, and ‘wayfinding’. This review encompasses publicly available documents, academic journals, conference papers, reports, and university theses from general academic engines, public scholar databases, and the university’s internal database up to the beginning of 2024;Efforts to Reduce Selection Bias and Identify All the Relevant Literature: Multiple reviewers assess the literature for reliability. A systematic search of the ‘grey literature’ is conducted using internet search engines;Quality Ranking of the Reviewed Literature: The literature with higher international impact and research significance is given higher importance;Suitability of Included Studies: Tables summarize the studies, providing information on authors, system performance, main system components, advantages, and disadvantages;Results and Interpretations: Findings are discussed at the end of each section, comparing them to other published works and related topics;Review Limitations: Limitations and future research directions are addressed in each section;Conclusion: The final section offers a concise summary of the primary review findings and the objectives of this review.

This paper aims to lay a robust foundation for the development of a fully autonomous positioning/navigation assistant system dedicated to VIPs. The objective is to overcome the limitations inherent in current mobility aids. Such a system would contribute to advancing the field and fostering the development of improved mobility aids for VIPs in complex environments.

## 2. Literature Review of Types of Vision Impairment

In order to develop a high-performance positioning system for navigation and wayfinding for VIPs, a prerequisite is to consider the details of the diverse visual impairment as classified by the International Classification of Diseases, 11th edition (ICD-11) [2], the UK Certificate of Vision Impairment (CVI), and pertinent international standards. It is worth noting at the onset that International Standards Organisation (ISO), is an independent, non-governmental, international standard development organization composed of representatives from the national standards organizations of member countries. Their headquarters are in Geneva, Switzerland. 

### 2.1. WHO Classification of Visual Impairment

In accordance with the World Health Organization (WHO)guidelines outlined in the ICD-11, Visual Impairment (VI) can be categorized based on distance and near acuity impairment and classified into various severity categories [2], as illustrated in Table 1. In studying visually impaired mobility, it is essential to establish consistent criteria for defining and categorizing the severity of VI to ensure comparability among studies using similar discretized experimental samples. The WHO (2018) [2] classification serves as the global standard, offering a robust and reliable reference framework.

However, the categorization introduced by the ICD-11 is primarily limited by its reliance on a fixed visual acuity as the key indicator of any VI level. In reality, visual acuity is inconsistent and varies across different operational scenarios. VIPs exhibit varying performance in different illumination levels; for instance, individuals with Hereditary Retinal Diseases (HRDs) often experience diminished visual function in low-light conditions [3]. Hence, it is important to assess a combination of visual functions in diverse environmental settings. Moreover, the ICD-11 may not incorporate new knowledge that contributes to the current comprehension of VI [4]. Beyond the universal ICD-11 standard [2], regional variations in VI classification may exist across different geographical areas [5].

### 2.2. UK Classification of Visual Impairment (CVI)

In the UK, the CVI is utilized, and it includes the following two categories of individuals:Sight impaired (partially sighted);Severely sight impaired (blind).

In 2018, the CVI incorporated self-reporting mechanisms concerning individual living circumstances. Furthermore, the revised CVI excluded both sight variation under different light levels and the comprehensive assessment of peripheral and central field loss [6].

The revised CVI in 2018 introduces increased ambiguity, with two primary limitations: (i) insufficient details regarding the severity of VI and (ii) discrepancies between the UK CVI and ICD-11. This has certain consequences. First, objective measures of visual function may not rigorously detail VI severity, and subjectivity from healthcare professionals can abrogate these measures. Second, misalignments with ICD-11 classifications highlight gaps between global and local standards. Ongoing revisions of the CVI continue to differentiate these standards in their classification methods [5].

### 2.3. Classification of Visual Impairment in International Research

The significance of employing consistent classification methods in research is essential, ensuring a robust comparison of the results. Ambiguities in international standards result in disparities in describing VI test samples, as illustrated in Table 2, which summarizes ten research works on visual impairment, mobility, and evacuation.

The main limitations of research outlined in Table 2 include (i) insufficient control for similar variables or an equivalent number of subdivisions, (ii) hindered experimental validity, (iii) lack of precise value, (iv) lack of consideration of participants’ general health conditions, (v) time-consuming self-report and natural history reports, and (vi) high costs for specialists to determine visual function accurately [5].

## 3. Literature Review of International Standards for Navigation for Visually Impaired People

In order to address the wayfinding and navigation requirements of VIPs, it is essential to explore the existing international and regional standards and frameworks governing VIP navigation. These standards can support the development of VIP travel aid designs that align with international benchmarks. Reviewing these standards serves as a solid foundation for conducting experiments aimed at understanding VIP behaviors in the real world.

### 3.1. ISO Standards of Navigation for Visually Impaired People

The key document is the International Organization of Standards (ISO) in 2020, ISO 21542 [17], entitled “Building Construction. Accessibility and usability of the built environment”. This provides guidelines for the design, construction, management, and maintenance of buildings, covering individuals with diverse ages, abilities, and behaviors in normal circumstances and emergencies. In particular, it encompasses requirements for the design and construction of building access, circulation, and evacuation. The ISO standards offer universal design principles to create environments that are accessible and usable for individuals with diverse disabilities, including those with visual impairments [17].

This section outlines the essential standards for VIP navigation. These include standards for tactile, visual, and audible information for VIP orientation, provision of emergency warnings, obstacle detection, and other relevant items.

Tactile information, such as the path of travel across large public areas, can be conveyed through facilities such as Tactile Walking Surface Indicators (TWSIs). As illustrated in Figure 2, the ISO standard outlines specifications for TWSIs, including dimensions for flat-topped elongated bars, spacing of ribs, and width at the base [17]. In addition, changes in materials at decision points (e.g., entrances/exits) can enhance the delivery of tactile information.

The effective delivery of visual information for VIP orientation relies on regulating variables such as luminance and contrast across various scenarios. Luminance and contrast adjustments along routes can assist VIPs in their navigation, while additional illumination and visual contrast can highlight decision points, such as staircases or doors. Table 3 presents three standards of minimum Light Reflectance Values (LRV) and contrast values for different visual tasks. Table 4 delineates standards of minimum lighting levels in different building areas [17]. Moreover, ISO standards emphasize the importance of designing both external and internal lighting to prevent reflection and glare, alongside the provision of legible signage (e.g., Braille) and graphical symbols to effectively assist with VIPs’ orientation.

The ISO standards highlight the criteria for delivering audible information (key destinations such as lifts, PoI) through audio transponders or beacons to facilitate VIP navigation. These include measures to manage noise (e.g., sound absorption or amplification), as well as hearing enhancement techniques (e.g., induction loop systems) in locations such as public transportation terminals.

In addition to fundamental tactile, visual, and auditory information, emergency warning information is crucial for ensuring the safety of VIPs during navigation. The ISO has established relevant standards for delivering audible emergency warnings through building alerting systems, visual emergency warning signals (e.g., fire alarms accompanied by visual alarms), and alerting systems in lifts. Furthermore, ISO has set complementary standards for emergencies, including areas of rescue assistance, the minimization of obstacles and hazards along pathways for VIPs using navigation aids such as white canes, and standards for relief facilities for assistance dogs.

The ISO standards provide a comprehensive guideline for all disabilities. However, they lack standards for external environments such as public open spaces, potentially causing discontinuous navigation through transition areas. As an international standard, it may not fully consider regional variations in VIPs’ navigation needs. Additionally, it does not fully address local regulations, leading to the need for extra compliance measures for VIP navigation in diverse countries.

Implementing all standards outlined in ISO presents complexity and significant expense for building developers, particularly for existing buildings. There is an absence of cost–benefit analyses and precise guidance on fostering collaboration and cooperation among architects, engineers, and stakeholders to implement ISO standards.

ISO standards fail to account for the rapid advancements in navigation aids for VIPs. Emerging technologies such as augmented reality, cloud-based systems, and smartphone-based navigation systems provide new opportunities for VIPs. Despite periodic revisions to ISO standards, the lengthy revision process hinders the timely integration of new technologies in developing navigation aids for VIPs.

Furthermore, the ISO standards may not fully incorporate the feedback from VIPs regarding their lived experience and navigation requirements under complex environments (e.g., urban areas and public transportation). Hence, there is a vital need to elicit relevant information from VIPs relating to their lived experience by using, for example, interviews and carefully designed experiments. This means that closely involving VIPs in the development of navigation systems will effectively meet their requirements.

### 3.2. Standards of Navigation for Visually Impaired People in International and Regional Institutes

It is noteworthy that various standards add value to ISO 21542 by providing additional improvements. For instance, ISO 9241-171: Ergonomics of human-system interaction provides guidance for designing software for interactive systems with high accessibility, thereby enhancing the usability of navigation systems for VIPs [18]. This standard provides numerous guidelines, including:(i)user-interface elements;(ii)user preference settings;(iii)accessibility adjustments;(iv)controls and operations, e.g., switching of input/output alternatives, optimization of the number of steps;(v)compatibility with assistive techniques, e.g., enabling concurrent operation of multiple assistive technologies.

Nevertheless, ISO 9241-171 fails to directly address the complex requirements and challenges continually faced by VIPs in physical environments. Moreover, certain standards, such as those for interface designs, may not be directly applicable to physical aids such as tactile maps. There is, therefore, an essential need for a comprehensive analysis of the integration requirements between software applications and physical aids as an initial step. This should be followed by the integration itself.

European Standard EN 301 549 offers comprehensive guidance on improving the accessibility of Information and Communication Technology (ICT) products and services (web-based, non-web, and hybrid) for diverse disability groups, including VIPs [19]. It delineates accessibility requirements for VIPs from digital information (e.g., auditory output delivery and correlation) to relay or emergency service (e.g., lip-reading relay service). Its standards of web construction for VIPs were enhanced by closely aligning with the Web Content Accessibility Guidelines (WCAGs). Similar to the constraints of ISO 21542, the digital-focused standards of EN 301 549 neglect considerations for physical aids such as tactile maps for VIPs and struggle to match the pace of evolving technologies for accessibility.

The 2010 Americans with Disabilities Act (ADA) Standards for Accessible Design is a revised version of the ADA in 1990. This standard is enforced by entities such as the Department of Justice and the Department of Transportation in the United States. It outlines both scoping and technical requirements for accessibility. They are to be implemented during the periods of design, construction, additions to, and alteration of sites, facilities, buildings, and elements as mandated by regulations issued by Federal agencies under the ADA of 1990 [20]. Notably, the 2010 ADA includes standards to enhance the navigation of VIPs, such as the removal of barriers at lift call buttons, speech-output-enabled machines for providing fully accessible and usable information, tactile signage, and accessible routes. Nevertheless, there are limitations in implementing this standard, especially as it is revised infrequently, e.g., the latest version dates back to 2010. It, therefore, does not incorporate recent navigation technology developments for VIPs. The 2010 ADA only sets minimum accessibility requirements for disabilities and lacks a detailed exploration of how navigation needs change with different levels of vision impairment. In addition, the impact of this standard is limited to the United States, highlighting the importance of considering additional international standards to ensure accessibility for VIPs worldwide.

British Standard (BS) 8300, comprising parts BS 8300-1 (external environments) [21] and BS 8300-2 (buildings) [22], provides comprehensive guidelines for designing accessible environments for VIPs in the UK. These standards cover external elements such as streets, parks, landscaped areas, approaches to buildings, and spaces between and around buildings to achieve an inclusive environment. BS 8300-1 emphasizes tactile pedestrian surfaces, appropriate lighting, audible signals at crossings and junctions, tactile maps, and high-contrast signage for navigation safety. BS 8300-2 addresses internal aspects, including accessible entrances with tactile signage, wide circulation areas with contrasting colors, handrails on both sides of stairs, contrasting nosings on steps, and braille signage. It also recommends best practices from consultations with VIPs and regular staff training. While implementing these standards promotes independence of mobility and compliance with the Equality Act, it also entails problems such as significant costs and complexity during construction.

### 3.3. Summary of International Standards for Navigation for Visually Impaired People

To summarize, both ISO standards and standards from other organizations demonstrate limitations because they are impractical to apply in certain cases and regions (Table 5). They have limited coverage of public environments while also presenting a slower update rate compared to rapidly evolving technologies. There is also insufficient incorporation of VIP feedback on real navigation needs. The ongoing revision of international standards requires guidance on collaboration among stakeholders, integration of emerging navigation aids for VIPs, and addressing real navigation requirements of VIPs with varying levels of VI under complex environments.

## 4. Literature Review of the Requirements of Visually Impaired People

The foundation for developing a positioning and navigation system for VIPs hinges on a thorough exploration of the requirements specific to any particular VI. This is indispensable for validating ongoing research and development efforts in this domain. Section 4.1 of this paper provides detailed definitions of mobility capacity and mobility performance. Furthermore, environmental and user requirements for specific applications are outlined in the same section. Section 4.2 delves into the details of system and user requirements.

### 4.1. Mobility Capacity, Mobility Performance, Environmental and User Requirements

Mobility capacity is the fundamental ability of individuals to mobilize. This is typically assessed in a controlled setting, often a laboratory, and this evaluation allows for the individual optimization of both physiological and psychological elements. Mobility performance is the realistic ability of an individual to mobilize, accounting for their motivation and considering the influence of the surrounding environment and mobility tools. Figure 3 illustrates the correlation between capacity and performance, elucidating their influence on behavioral modeling. The comprehension of mobility capacity and performance evolves through an iterative methodology, where an individual’s motivation, environment, and mobility tools are systematically isolated, tested, and refined [5].

In the development of mobility performance, as is evident in several studies [23,24], the fundamental requirements of VIPs encompass the following:Obstacle and hazard awareness;Orientation and wayfinding (‘Where am I?’).

Prominent intervention methods to enhance mobility performance involve environmental augmentations and the use of mobility aids. The optimization of specific environmental augmentations has been shown to improve mobility performance. These include factors such as illuminance level and color temperature, contrast, color, signage, and auditory, olfactory, and tactile interactions. A research gap is evident, emphasizing the need for future studies to establish specific standards for color temperature, hue, and chroma. Furthermore, exploring their consequences on perceived contrast and objectively measuring mobility performance is crucial for advancing our understanding in this domain [5].

Table 6 outlines the requirements for general environmental features. A literature gap is denoted as a concept yet to be explored in existing research. The significance of these gaps lies in their potential impact on mobility performance. Requirements are categorized as either ‘hard’ (essential) or ‘soft’ (desirable or ideal), aligning with the framework proposed by [23]. These specifications delineate ways in which navigation systems can address current gaps, which are contributed by [25].

### 4.2. System and User Requirements

Required Navigation Performance (RNP) of positioning, which includes the following: accuracy, integrity, availability, and continuity, is essential for navigation applications such as VIP navigation assistant systems. By providing information with centimeter positioning accuracy in real-time, PNT sensors, such as GNSS, can effectively achieve RNP. For instance, implementing GNSS in mapping and Geographic Information System (GIS) applications expedites the precise data acquisition process, leading to reductions in equipment and labor expenditures [27].

In addition, PNT technologies can provide precise positioning by determining the user’s location on Earth. These technologies can utilize a user’s positioning/timing information to obtain routes and turning instructions. Such PNT technologies can also enable precise time information, thereby helping to synchronize different parts of the navigation system. This ensures the provision of timely information during a VIP’s navigation. Furthermore, technologies are transferable; the knowledge and experience from other PNT applications can support developing the navigation system for VIPs to navigate independently and confidently.

In conjunction with environmental enhancements in Section 4.1, mobility tools with PNT technologies have been devised to aid VIPs. Study [28] categorizes current mobility tools into three distinct types:Electronic Travel Aids (ETAs), designed for obstacle detection and hazard awareness;Electronic Orientation Aids (EOAs), focused on orientation and wayfinding (‘Where am I?’);Binary Electronic Mobility Systems (BEMS) seeks to combine the advantages of both ETAs and EOAs.

These mobility tools have inherent limitations. Early devices predominantly comprised ETAs that struggled to address challenges beyond obstacle detection for a VIP, neglecting their orientation and navigation needs. While the introduction of EOAs aimed to rectify this deficiency, nevertheless, the positioning technology underlying current EOAs falls short in providing comprehensive indoor and outdoor coverage, confining users to a combination of ETAs and EOAs based on the operational scenario. Beyond the conventional ETAs, EOAs, and BEMS, an emergent technology, Sensory Substitution (SS), captures the environment visually and translates it into an alternative sensory input, unlocking the potential to convey a dense array of information to the user. However, a limitation of SS technology lies in its inability to deliver this information density in a medium that can be processed by the user. In summary, for these mobility tools to be effective, they must tackle two major challenges: 1. Effectively categorizing information that is useful to the user. 2. Delivering this information to the user in an accessible and usable format [5].

The deficiencies in these technologies can be attributed to the segregation of visually impaired research between the medical and engineering disciplines, which, as a consequence, has impeded technology developers from accessing a comprehensive set of user requirements.

The following outlines the key user requirements for mobility tools designed for visually impaired individuals within the current positioning domain:Smaller Size and Lightweight (higher scalability): Navigation systems with considerable dimensions and weight hinder their adoption by VIPs for navigation purposes;Higher Affordability (cost-effectiveness) and Lower Learning Time: The substantial cost and learning time associated with existing systems discourage users from dedicating significant energy to familiarising themselves with the navigation technology;Less Infrastructure Implementation: Navigation systems requiring significant environmental changes, such as the installation of BLE beacons at Points of Interest, pose challenges and entail additional infrastructure investments;Multimodal Feedback Design: Many assistant aids incorporate only an audio feedback system, which may prove ineffective in noisy environments. Given the critical role of the feedback system in navigation, it should be designed to offer multi-modal feedback options;Real-time Information Delivery: The complexity of object detection operations results in delays in real-time information delivery. Any delay poses a risk of exposing users to hazardous situations;Privacy Considerations: Digital systems may put VIPs at risk of privacy leaks. None of the mentioned systems have adequately addressed data management, including audio and image data, during and after navigation. Establishing ethical professional standards for system development is crucial to safeguarding users’ data [29];Coverage Area/Limitations: Assessing the system’s range from single-room to global capabilities, considering specific environmental limitations that might impact visually impaired users;Market Maturity: Examining the development stage of assistive tools for VIPs, from concept to product availability, to determine their market maturity;Output Data: Evaluating the types of information provided by the system, including 2D or 3D coordinates, relative or absolute positions, and dynamic parameters such as speed, heading, uncertainty, and variances [30];Update Rate: Determining how frequently the system updates information, whether on-event, on request, or at periodic intervals, to meet the needs of VIPs [30];Interface: Considering the various interaction interfaces, including man-machine interfaces such as text-based, graphical, and audio, as well as electrical interfaces such as USB, fiber channels, or wireless communications, for optimal accessibility [30];Scalability: Assessing the scalability of the system, taking into account its adaptability with area-proportional node deployment and any potential accuracy loss;Approval: Considering the legal and regulatory aspects regarding the system’s operation, including its certification by relevant authorities, to ensure compliance with standards and enhance user trust;Intrusiveness/User Acceptance: Evaluating the impact of the system on VIPs, distinguishing between disturbing and imperceptible levels of intrusiveness to ensure high user acceptance;Number of Users: Determining the system’s capacity for the visually impaired group, ranging from single-user setups (e.g., total station) to support an unlimited number of users (e.g., passive mobile sensors), ensuring inclusivity [30].Long-term Performance: Integrating longitudinal studies to assess navigation systems’ usability and effectiveness for VIPs is crucial for understanding their adaptation to changing conditions and ensuring sustainability.

It is evident that a notable research gap exists in quantifying these requirements to a significant extent, given the multitude of criteria; it is challenging for users to identify an optimal system for a specific application. All of the above requirements show the complexity and multidimensionality of the optimization problem faced by users. For each application, the sixteen user requirements above need to be carefully weighed against one another [30]. To meet market demands, any embedded technology must exhibit characteristics such as being cost-effective, energy-efficient, low-latency, compact, requiring minimal maintenance, involving a minimal amount of dedicated infrastructure, and meeting the requirements for RNP across diverse and complex environments.

To bridge the research gap arising from the lack of behavioral data to understand the requirements of VIPs in realistic scenarios, Section 5 explores experimentation at the Pedestrian Accessibility Movement Environment Laboratory (PAMELA) to overcome this. This experimentation aims to investigate and compare the behavior of VIPs both in individual settings and within crowds. The section concludes by consolidating the findings and offering future recommendations derived from the experimentation. This contribution serves to refine the requirements of VIPs for navigation and wayfinding in realistic scenarios.

## 5. PAMELA Experimentation for Investigating and Comparing the Behavior of Visually Impaired People

This section outlines experimentation conducted to bridge the literature gaps outlined, encompassing the absence of behavioral data for VIPs in individual and crowd scenarios. It addresses the necessity for an empirical and consistent experimental process to effectively overcome the gaps, highlighting the imperative for an empirical and consistently applied experimental process to address this gap.

### 5.1. Overview of PAMELA Experimentation

The PAMELA (Pedestrian Accessibility Movement Environment Laboratory) experiment aims to observe visually impaired participants in controlled environments, analyzing their behavior amidst unidirectional and opposing crowd flows. The goal is to develop an empirical model predicting the movement of visually impaired people in diverse scenarios, both individually and within crowds, including normal and evacuation situations. The findings from this experiment serve as a basis for subsequent research, contributing to the refinement and expansion of environmental features and indoor navigation systems [5].

PAMELA serves as a versatile platform for the investigation of pedestrian mobility and environmental interactions under controlled conditions. The facility’s variable lighting system, capable of adjusting illumination levels from near darkness to over 1000 lux, allows for the exploration of the impacts of lighting on mobility. The platform layout, validated for empirical, repeatable, and real-world relevance, consists of three defined sections: (1) a straight path, (2) a maze with barriers, and (3) a straight path with two foam blocks at ground level, 1.2 m in length, 0.2 m in depth, and 0.13 m in height (Figure 4). Lighting, calibrated to a uniform 256 lux at platform level, simulates typical indoor lighting. Cameras positioned over each section record all aspects of the experiment, while Pozyx, a Real-time Locating System (RTLS), tracks participants. Codamotion tracks participants in one section, enabling a separate data fusion study. Video footage, Pozyx, and Codamotion data are cross-compared for reliability, and questionnaires complement data collection. The platform’s controlled environment and integrated systems provide a comprehensive framework for investigating pedestrian behavior and mobility.

The testing scenarios included the following five different environments:individual;unidirectional group flow restricted;unidirectional group flow unrestricted;opposing group flow restricted;opposing group flow unrestricted.

In the individual scenario, participants completed one forward and one reverse pass for each section. For all other scenarios, participants performed one pass per section in a restricted and unrestricted group flow. Restricted scenarios required participants to adhere to the speed of the slowest person, fostering group cohesion, while unrestricted scenarios allowed free traversal without jogging or running, inducing higher stress and dynamic behaviors. Unidirectional group flow involved VIPs in groups with Normally Sighted People (NSPs) moving in one direction. Opposing group flow tested two groups simultaneously, each with up to seven NSPs and one VIP, moving toward each other in opposing directions. Data collected included the following: time to traverse sections, effective speed, qualitative behavior from video footage, walking speed, walking path, and psychological performance from questionnaires.

In participant recruitment, VIPs were classed on the basis of a formal diagnosis of VI beyond correctable refractive errors, supported by a CVI, while NSPs did not have VI beyond correctable refractive errors. Separate questionnaires, detailed in Appendix A and Appendix A, were designed for VIPs and NSPs, covering age, gender, and previous evacuation experience. VIP questionnaires additionally gathered information on VI, and post-experiment sections assessed participant comfort on each platform section using a 5-point Likert scale. These experiments, conducted between 9 and 11 April 2019, involved 61 participants, including 12 VIPs with varied impairments. The experiment has limitations, notably in the uneven age distribution among VIPs, with 10 out of 12 individuals being above 50 years old. While this may present a constraint, it is important to emphasize that the primary factor influencing VIP mobility is the cause of VI, as supported by [31]. To maintain a focus on the diverse impact of distinct visual impairments, this study intentionally excluded individuals who self-reported the onset of age-related mobility problems.

The visually impaired group comprised nine males and three females, exhibiting an uneven age distribution. Table 7 outlines the self-reported visual function ratings (VA: Visual Acuity, CS: Contrast sensitivity, VF: Visual Field), aimed at identifying the most valued visual function, and explores correlations with mobility using a 3-point Likert scale (where 1 is poor, 2 is fair, and 3 is good). In contrast, NSPs had a younger mean age than VIPs in this experiment, potentially limiting the representation of the broader age range in realistic environments. Nevertheless, since the primary role of NSPs is to serve as dynamic obstacles with minimal cognitive interaction, the age difference is not considered to limit this research’s outcomes. Additionally, the older VIPs sample aligns with the higher likelihood of visual impairment occurring with age [5].

### 5.2. Summary of Performance of PAMELA Experimentation

Figure 5 shows the average effective speed for all VIPs (excluding H1 for a fair comparison) during their traversal of the platform in individual, unidirectional, and opposing flows. When considering the section average for each scenario, the effective speeds of VIPs exhibit a remarkable degree of consistency, demonstrating only minor variations (within a range of 0.2 m/s). Ordered in terms of decreasing velocity, the unrestricted unidirectional flow records the highest average effective speed of 0.9 m/s, followed by individual performance, restricted unidirectional, and unrestricted opposing flows (0.8 m/s). Ultimately, the restricted opposing flow presents the lowest average effective speed across sections, measuring at 0.7 m/s.

A major conclusion that can be inferred from these results is that group conditions failed to yield significant disparities in the overall effective performance of VIPs in comparison to their individual performance. Notably, the unrestricted unidirectional flow results in average performance enhancements for the majority of VIPs, in contrast to expectations based on the literature review. Additionally, Section 1 consistently exhibited the most favorable average effective performance, while, except for individual performance, Section 2 presented the least favorable outcomes. This implies that obstacles posed constraints on performance in all scenarios, with larger obstacles (including environmental boundaries) exerting a more substantial limiting effect on group performance when compared with smaller ground-level obstacles.

Table 8 illustrates the distribution of results for the average effective speed of all VIPs (excluding H1) during their traversal of the platform in individual, unidirectional, and opposing flows. The unrestricted unidirectional flow exhibited the smallest spread across sections, followed by the restricted unidirectional and unrestricted opposing flows. Notably, the unrestricted unidirectional flow possessed the largest and most consistent advantage in average effective performance relative to the other scenarios.

Figure 6 illustrates the average effective walking speed across sections for each VIP in different scenarios (ascending visual function). The VIPs were initially grouped and further sorted based on their self-reported visual function scores within each group. Notably, the visual function scores were discretely categorized into four unevenly sized groups (where Group 1 represents the lowest visual function, and Group 4 represents the highest visual function). This prevents a quantitative assessment of the observed positive trend in effective walking speed with increasing visual function. Additionally, the exclusion of opposing flow data from H1 in calculating their average further limits the available data points for defining this relationship. A more in-depth analysis of each group’s overall average supports the apparent positive trend, with Group 1 exhibiting the lowest average effective speed at 0.6 m/s, followed by Group 2 at 0.7 m/s, Group 3 (excluding H1) at 0.9 m/s, and Group 4 recording the highest value at 1.2 m/s. Despite these trends, the spread of these data does not reveal significant correlations with increasing visual function, most likely because of the limited sample size within each group.

VIPs were interviewed to determine whether the presence of NSPs affected their behavior. Out of the twelve VIPs surveyed, nine indicated that the presence of NSPs influenced their behavior, with the majority of respondents (seven out of nine) asserting that such influence positively contributed to the overall performance of VIPs. Contrary to expectations, these findings suggest that interactions with NSPs are likely to enhance the performance of VIPs.

It is noteworthy that effective performance within the opposing flows did not solely depend on immediate interactions between opposing groups. Instead, it appears that each VIP may have modulated their walking speed from the beginning of the opposing pass. An inherent limitation lies in the lack of control over the presence of another VIP in every session due to constraints in participant recruitment. Future research should aim to explore opposing flows with a controlled number of VIPs. This is relevant for studies in environments such as hospitals, care homes, or other spaces where VIPs are more prevalent [5].

### 5.3. Conclusions of PAMELA Experimentation

Understanding VIPs’ requirements involves analyzing their behavior in various scenarios. In group settings, VIPs performed better than in individual scenarios. The unrestricted unidirectional condition showed the highest effectiveness, with 75% of VIPs experiencing improved performance. Conversely, the restricted opposing flow yielded the worst average performance, with only 45% of VIPs showing improvement.

Despite the effectiveness of unrestricted conditions, they led to undesirable behaviors, such as overtaking, collisions, and disorientation. In contrast, the restricted condition provided smoother flows, reflecting the preference of VIPs for enhanced safety and ease of lane formation. In all group scenarios, VIPs utilized the NSPs around them for orientation through visual, auditory, or haptic coupling, enhancing walking efficiency and obstacle avoidance. Unidirectional flows supported this behavior, while opposing flows led to collisions and deadlock.

A positive correlation was found between walking speed and self-reported comfort when VIPs traversed the platform individually. This correlation was maintained in unidirectional flows but diminished in opposing flows. In undesirable group conditions, VIPs’ control of walking behavior is coupled with the crowd’s state while they are decoupled from their psychological state.

NSPs prioritized contact avoidance with VIPs, enabling natural lane formations, particularly in opposing conditions. Removing VIPs from group scenarios resulted in significant reductions in NSP completion times but did not enhance flow safety. Furthermore, NSPs showed reduced effective performance in restricted versus unrestricted conditions. In restricted flows, NSPs preferred a line-abreast group formation, differing from the river-like movement in unrestricted flows, characterized by elevated and varied walking speeds, as defined by [32]. In opposing flows, the inadequate maintenance of group formations resulted in collisions in restricted passes, while collisions occurred at higher speeds in unrestricted passes. Both the NSPs and VIPs expressed a preference for the restricted condition.

In the presence of VIPs, the narrowest section (Section 1) showed the best VIP performance, extending to NSPs in restricted conditions. Removing the VIPs led to the widest section (Section 3), yielding the best NSPs performance. This underscores the VIPs’ impact on flow limitations, particularly in sections with obstacles (Section 2 and Section 3). The classical speed-density relationship applies to NSPs without a VIP but is inapplicable when a VIP is introduced [5].

### 5.4. Limitations and Recommendations of PAMELA Experimentation

The experiments conducted offer several noteworthy contributions. First, they contribute by carefully examining variations in behavior and performance among individual VIPs when operating independently and within controlled group settings. Second, this study significantly addresses a research gap, as existing models insufficiently associate a static reduction in walking speed to simulate a VIP’s behavior relative to their normally sighted counterpart. Third, it is established that models should consider individual, unidirectional, and opposing flows (including both restricted and unrestricted conditions) due to substantial variations in group behavior among them. Fourth, the experiments reinforce findings from the existing literature. For example, the classical inverse relationship between speed and density, as proposed by [33], persists in NSP-only group conditions. Consequently, while the novel behavior of VIPs is described, realistic NSP performance, as simulated in previous studies, enhances the credibility of the current experiment [5].

There are several limitations to this series of experiments. First, while the experiment has extracted novel VIP–NSP group behaviors, it has not been performed exhaustively, considering the multitude of visual impairment and operational scenarios that exist but could not be adequately represented in this study. The logistical constraints in participant recruitment and clinical visual function testing also restricted the depth to which correlations between speed, density, and visual function could be explored. Nonetheless, the diversity of participants’ visual impairment, ranging from no light sensitivity to moderate partial sight in both eyes, provided a sufficiently comprehensive performance dataset. The inclusion of VIPs with a diverse spectrum of walking speeds allowed for distinct characteristics in each pass.

Second, it is recommended that future research explore larger group sizes, as the maximum unidirectional group size of seven (and fourteen in opposing flows) may be inadequate in generating realistic densities in all scenarios. VIPs’ behavior predominantly constrained group performance, and the threshold at which density became the limiting factor was undiscovered because of the restricted group size.

Finally, the instructions provided to participants may need to be refined. Some participants did not consistently interpret the ‘restricted’ or ‘unrestricted’ instruction. For instance, some understood an unrestricted pass as an opportunity to traverse the platform as quickly as possible, while others interpreted it as an instruction to move freely. This misunderstanding led to inconsistencies in NSP–VIP interactions, with some participants overtaking the VIP while others remained behind, physically guiding the VIP in such instances.

For future experiments on VIP user requirements, it is essential to provide participants with a more comprehensive briefing to ensure their equal understanding of the experiment’s objectives in both unrestricted and restricted conditions. However, it is worth noting that inconsistent VIP–NSP interactions remain realistic due to participants’ individual interpretations of flow conditions [5], as observed in instances where group members assisted each other in evacuations involving VIPs [8].

Subsequent research should build upon the empirical foundations of the PAMELA experimentation to determine their manifestations in real-world scenarios. For example, the PAMELA experimentation did not address performance variations in scenarios such as varying angles of opposing flows (such as the meeting of opposing groups at right-angled intersections) or with environmental changes and different mobility aids, such as guide dogs [5]. In conclusion, future research will validate the proposed PAMELA systems across diverse real-world environments to ensure their practicality and effectiveness.

## 6. Literature Review of Current Positioning/Navigation Systems for Visually Impaired People

To achieve safe and efficient navigation for VIPs, it is essential to acquire non-visual environmental information through appropriate sensors. Overcoming challenges such as objects obstructing paths, suspended obstacles, traffic junctions, and slippery roads is crucial.

Conventional navigation aids include white canes, guide dogs, tactile paving, and trained volunteers. The most common aid chosen by VIPs is a white cane, a rod typically 1.0 to 1.5 m in length, which provides tactile feedback that extends a user’s perceptual range for obstacle detections (in 2D). The white cane can provide fundamental information for any user to achieve basic mobility. However, it cannot detect any obstacle above the user’s waist, leading to a danger of collisions. Another limitation is that users must continuously scan the environments through constant movements, incurring effort in daily life. A final drawback is that the white cane only provides limited detection distance, consequently limiting the time available for VIPs to undertake preventive actions.

Assistive tools and navigation technologies are becoming increasingly important in the lives of VIPs. They are created to support people with disabilities in overcoming physical, social, infrastructural, and accessibility barriers. The main contributions of VIP assistive systems are accuracy, reliability, continuity, wayfinding, safety of navigation, interaction, and information access. This section reviews the positioning accuracy and limitations of recent systems of the following three categories of assistive systems that support VIP navigation:(i)vision-based navigation systems that utilize vision sensors (e.g., cameras) to detect obstacles and ensure safe navigation for VIPs;(ii)non-visual sensor-based navigation systems that guide VIPs, such as Bluetooth beacons, infrared, ultrasonic, maps, sound, and smartphones;(iii)smart systems based on technologies such as artificial intelligence and machine learning.

### 6.1. Visually Impaired Navigation with Vision Systems

Vision systems use sensors (with computer vision algorithms), such as cameras, to extract visual features from the environment. This type of navigation system detects obstacles through visual features. It avoids collisions by employing advanced pathway planning and feedback mechanisms, thereby ensuring the safety of VIPs. Comprehensive details of vision systems are provided in Table 9.

Ref. [34] designed wearable E-Glasses, which include an RGB-D camera, an embedded computer, headphones, and a microphone. The glasses incorporate a target detection network to fuse visual and auditory information, thereby assisting VIPs in localizing targets. Additionally, the glasses incorporate a neural path planning system that combines spiking neural and convolutional neural networks. These efforts resulted in high success rates of 95.46% in target detection and 92.60% in optimal path prediction. Nevertheless, there are areas for improvement:(i)enhancing battery life and exploring alternative power sources;(ii)improving E-glass design for comfort and aesthetics;(iii)further investigation of the real-world requirements of VIPs, particularly in complex outdoor environments.

Ref. [35] developed the Intelligent Situational Awareness and Navigation Aid (ISANA), which uses an RGB-D camera and Google Tango (as the mobile computing platform). ISANA can achieve a localization error smaller than 1 m without drifts (closed loop). It can achieve obstacle detection and avoidance using the Kalman filtering algorithm. In addition, the system is enhanced by the multimodal Human–Machine Interface (HMI): speed-audio and robust haptic interactions (by electronic smart cane). It is worth noting the following limitations of ISANA:(i)there is a need to improve the user interface, such as in better speech recognition to achieve robustness in noisy places;(ii)the audio frequency should be adjustable and customized. For example, the system cannot navigate in complex, cluttered environments such as underground stations.

Ref. [36] proposed a novel assistive system for VIPs, focusing on obstruction avoidance based on a Raspberry Pi camera. The camera is assembled on a stick and continuously captures real-time pictures. The pictures, along with estimated data, undergo analysis in the Raspberry Pi board. The system achieves high object detection rates with an average efficiency of 91%. The principal limitation of this system, though, is the distortion within the captured images.

Ref. [37], for example, developed a vision system that integrates the binocular camera, Inertial Measurement Unit (IMU), and earphones on a helmet. The sensing range of this system is between 10 and 20 m. A sound is conveyed to the earphones when an object is detected, enabling users to walk in unknown urban areas. This system applies Binaural rendering, a technology for creating sounds that can be localized in directions and distance (with headphones). The system demonstrates environmental perception capabilities through geometric scene modeling using the binocular camera. Additionally, it incorporates a robust head tracking algorithm, combining data from the IMU and visual odometry to achieve precise and low-latency head pose estimation, encompassing both orientation and position. Nevertheless, the system exhibits the following limitations:(i)the system has yet to operate in indoor environments;(ii)although sonification can notify users about dangerous objects, the lack of decision on a direction to walk will increase the risks of danger;(iii)the dimensions and weights of the bulky helmet and backpack must be minimized to address concerns of portability.

Ref. [38] applied the Visual Simultaneous Localization and Mapping (VSLAM) algorithm. The average deviation distance is less than 0.3 m with a small variance. The algorithm includes a dynamic sub-goal selection method to avoid obstacles. The system can attain the required accuracy for route following and obstacle avoidance, but it shows insufficient continuity because of the reliance on cameras (e.g., worse performance at nighttime and in extreme weather).

LiDAR Assist Spatial Sensing (LASS) is a system proposed by [39] and performs better in detecting closer obstacles (because of a higher obstacle hit rate) and obstacles at a 90-degree angle. LASS uses a LiDAR sensor to detect obstacles and can convert the message to sounds of different pitches. A major advantage of LiDAR is that it can provide spatial information, such as the orientation of the obstacle. The distance between the user and the obstacle is translated into relative pitches. However, using the laser pointer to point at targets is difficult, and hence, considerable training time is needed for users. Furthermore, the system requires a significant length of time to perform full scans. Applying multi-sensors is recommended when facing a complex layout of obstacles in real-world environments.

### 6.2. Visually Impaired Navigation without Vision Systems

This section discusses navigation systems that use sensors other than the vision algorithms or sensors. These systems offer VIP navigation guidance based on non-visual attributes, i.e., they use sensors or devices encompassing the following: Bluetooth beacons, infrared, ultrasonic, maps, sound, and smartphones. Detailed information about the systems is presented in Table 10.

Ref. [40] proposed an indoor navigation solution that integrates Bluetooth Low Energy (BLE) beacons and Google Tango. This hybrid system requires less reliance on volunteer interventions than the BLE-only navigation system, but the travel duration is similar. Tango has an RGB-D camera and can provide highly accurate locations. Its feature map is limited to one floor each time. BLE beacons can provide excellent detection of coarse location, thus can find which Area Description File (ADF) Tango can load to report accurate position. This system effectively reduces smartphone power consumption. Nonetheless, BLE signals are subject to considerable noise because of the attenuation of internal building materials and are incapable of attaining high positioning accuracy.

Ref. [41] proposed an indoor navigation system called SUGAR, which uses Ultra-wideband (UWB) for effective positioning. The magnitude of positioning errors is up to 20 cm in most instances. Compared with other RF techniques, UWB uses a narrower radio frequency pulse to distinguish direct-path signals (carry useful information) from reflected signals. The presence of obstacles significantly affects the Received Signal Strength Indicator (RSSI). However, UWB technology combines the position estimation method (with time or angle of arrival) to mitigate the impact of obstacles during localization. In addition, systems based on UWB have greater transmission ranges than other techniques, such as Radio Frequency Identification (RFID), and require fewer devices. SUGAR can find pathways for VIPs using the A* algorithm, which is primarily used to find the shortest path between two points. The SUGAR system can interact with users through acoustic signals and voice through headphones. While this system demonstrates its suitability for deployments in public events, including host references and meetings, it requires considerable expense for small-scale deployments, circumscribing its adoption for private residences.

Ref. [42] implemented Radio Frequency Identification (RFID) techniques. The RFID tags are fixed in the main points along the path. The system utilizes GPS or the nearest RFID tag to detect locations and guide VIPs to their destinations by reading the next detected RFID tags in the pathway. The smart system shows proficiency in obstacle avoidance. It exhibits limitations in detecting moving obstacles and suspended objects.

Ref. [43] created a non-vision system named Navguide, which uses six ultrasonic sensors to categorize obstacles and environments. Under most experiment sets (indoor, outdoor, a U-shaped corridor, and an ascending staircase), Navguide reduces the rate of collision with obstacles. Participants with Navguide can complete the tasks more efficiently than those solely using white canes. Navguide has certain notable advantages: (i) it is cost-effective and lightweight, (ii) it can provide simple and priority information to users by vibrations and audio, and (iii) it shows proficiency in detecting wet floors (mitigating slipping accidents), floor-level obstacles, and obstacles below users’ knees. However, limitations exist within the system: (i) it cannot detect descending stairs, thereby elevating the risk of falls; (ii) it only senses when users step on the wet floor, and there are limitations in sensing pits.

Ref. [44] developed the Indoor Navigation Assistant for Visually Impaired (INAVI), utilizing public building Wi-Fi networks for navigation. The system employs a NodeMCU ESP8266 device to scan for Wi-Fi signals and determine their RSSI values. Trilateration is performed using the three strongest Wi-Fi signals, enabling user localization. Speech output guides visually impaired users, facilitating independent navigation within buildings. INAVI demonstrates a 96% accuracy in location recognition and an average localization accuracy of 1.5 m. Its high portability is facilitated by minimal hardware (only ESP8266 and a speaker). Limitations include potential delays in signal reception, signal fading affecting RSSIs, and the absence of obstacle guidance.

Ref. [45] designed a wearable haptic system in vibrotactile guidance shoes for VIP navigation. The system incorporates an Electronic Orientation Aid to process GPS data through GIS, a Bluetooth connection for sending guidance to the shoes, and vibrator motors to transmit directions to VIPs (e.g., vibrator 1 for forward direction). The experimentation demonstrated an average accuracy rate of 94% in identifying four main directions (e.g., up, forward, right, and left), facilitating successful wayfinding of proposed paths by all participants. Nevertheless, there are notable areas for enhancement: (i) users faced challenges in recognizing up-right and up-left directions, (ii) multiple outages occurred during testing, and (iii) the performance under diverse environmental and ground conditions remains unexplored.

Smartphone-based navigation systems provide convenience and portability to VIPs. NavCog3 [46] is an indoor smartphone-based navigation system that provides a position with an average accuracy of 1.65 m. During the experiment to test NavCog3, 93.8% of 260 turns were successful, but it is more difficult to turn by 45 degrees than 90 degrees. NavCog3 can provide turning instructions to users, and the user receives instant audio feedback when their orientation is incorrect. The system can also provide essential information such as distance, location, and direction of landmarks and Points of Interest (PoIs). Notably, NavCog3 relies primarily on smartphones as its core component, obviating the need for additional signal receivers. Nevertheless, NavCog3 has the following limitations: (i) for large-scale environments, the deployment cost is high, and it is time-consuming to collect landmark information indoors, and (ii) lack of control of users’ exposure to earlier versions of the system, which can affect the accuracy or subjective responses.

Infrared (IR) sensors present notable advantages, such as lower power consumption and cost-effectiveness compared with visual sensors. For example, [48] developed a navigation system for VIPs using infrared sensors. To compensate for the deflection of the system, this navigation system integrates gyroscope, accelerometer, and magnetometer sensors. The sensors are placed on users’ arms, and the transmitted signals can provide information on movement steps and the closest obstacles or threats. The system achieves an average accuracy of 20 cm and takes advantage of temperature differences to detect objects such as walls. However, the positioning accuracy depends on the resolution of the infrared sensors, highlighting the need for further enhancements to ensure the system’s effectiveness in navigating complex urban environments.

### 6.3. Visually Impaired Navigation with Smart Technologies (Artificial Intelligence)

In recent years, there have been an increasing number of advancements in real-time navigation technologies, including Artificial Intelligence (AI), machine learning, computer vision, and smartphone-based technology. These innovations have led to the development of more intelligent systems than white canes to assist VIPs in their daily navigation. The main types of navigation assistance for VIPs include smart canes, smart glasses, and smartphone-based systems. These smart navigation aids provide users with detailed information relating to their surroundings, enhancing the independence and mobility of VIPs [48].

Smart canes, in particular, offer significant improvements over traditional white canes by enabling the detection of obstacles at various heights. They extend the detection range from the knee level up to the eye level, thereby providing a more comprehensive environmental awareness for users than traditional white canes [49]. Smart canes [50,51] show advantages such as ease of learning and use with minimum external assistance. Smart canes can be enhanced using the Internet of Things (IoT) to interconnect artificial vision, GPS, and eye rings [52]. Ref. [53] designs a system of a wearable smart belt on the waist and a smart stick. The system is affordable and does not require an Internet connection. However, similar to smartphones, first-time users of smart canes need a period of adjustment prior to proficiency. Smart canes are limited by their reliance on an Internet connection and GPS-GSM module [54], and require detailed testing and experimental processes to validate their efficacy [55]. They also have low compatibility with different levels of visual impairment where a cane is not required [56].

Smart glasses [57,58] are another navigation aid suitable for VIPs, including those with physical disabilities. They offer convenience for daily use. However, smart glasses cost more than traditional white canes, making them inaccessible to middle- and low-income families [49].

AI-based smartphone navigation assistants present a cost-effective alternative. These systems can provide detailed information about obstacles and environments without the need for external data networks or substantial infrastructure [59]. Smartphone-based systems [60] are easy for VIPs to hold or wear. The navigation functions achieved by smart systems are shown in Table 11 [49]. Nonetheless, there are issues with these systems that must be addressed. Users may have to make the camera face forward while navigating, which can cause inconvenience. In addition, the requirement of electrical power remains a challenge for any journey of significant duration. Recently, several studies have explored the implementation of deep learning in assistive systems, though their testing has primarily used historical data. This highlights a significant research gap, as there is a need to conduct experiments with these systems in diverse, real-world environments to validate their effectiveness [61].

Artificial Intelligence plays a vital role in object detection. There are two categories of object detection methods: two-stage and one-stage methods. The former method obtains candidate regions and classifies them, while the one-stage method directly classifies and obtains positions. The most widely used two-stage methods are Region Convolutional Neural Network (RCNN) [62] and Region Fully Convolutional Network (R-FCN) [63]. Popular one-stage methods are the Single Shot MultiBox Detector (SSD) [64] and the You Only Look Once (YOLO) [65]. These two methods have differing qualities: the two-stage method excels in accuracy, while the one-stage method offers higher speed [61].

**Table 11 sensors-24-07020-t011:** A summary of innovative functions of smart navigation systems [51].

System Proposed By	Data Network Undependability	Coverage (Indoor/ Outdoor)	Obstacle Recognition	Distance Estimation	Position Estimation	Scene Recognition	Motion Detection	Multimodal Output
[66]	✓	✓	✓	✓				
[67]	✓	✓	✓					
[68]	✓	✓	✓	✓				
[69]	✓	✓						
[70]	✓	✓	✓					✓
[58]		✓	✓			✓		
[57]		✓	✓					
[60]	✓							
[49]	✓	✓	✓	✓	✓	✓	✓	✓

✓ Represents a feature or trait that is possessed by the system.

Ref. [61] proposes a mobile intelligent guidance system for VIPs. In such a system, phone cameras video the real-world tactile paving; then, this information is processed, and VIPs are guided by sound or vibration in real-time. This system achieves an image procession speed of nearly 14 FPS and a model accuracy of 93.76% [61]. The key intelligent technologies used are:(i)MobileNet-V2 model (fine-tuned by transfer learning) to extract features on overlapping grids;(ii)Single Shot MultiBox Detector (SSD) to detect TWSI;(iii)Score Voting algorithm to determine user’s positions.

Even though Convolutional Neural Networks (CNNs) achieve high performance in object detection, they suffer from significant costs because of model complexity [61]. The MobileNet-V2 model balances accuracy and latency in mobile devices by reducing model sizes and computation complexity. When tested on performance, the system shows a reduced latency at 60–80 ms in subway stations. At outdoor sidewalks, the system’s inference time for an image is 71 ms (14 FPS), which can satisfy real-time navigation requirements [61].

Ref. [49] proposes a deep learning smartphone-based navigation assistant system named DeepNAVI. This system can provide information on the types of obstacles, their positions and distance from users, motion status (stationary or moving), and scenes. The main components of DeepNAVI are a smartphone and a bone-conduction headset. Figure 7 shows the components of the DeepNAVI navigation assistant systems.

In addition, DeepNAVI has six software modules with high-performance models: obstacle detection (a lightweight model, EfficientDet-Lite4), position estimation, distance estimation, motion detection, scene recognition (SceneRecog model [71]), and output.

DeepNAVI includes an Android application that processes the trained obstacle detection and scene recognition models with the other modules. An Intel Xeon processor (with 64 GB RAM) and an NVIDIA GPU (GeForce GTX 1080 Ti) are used to train the deep-learning models for obstacle detection and scene recognition. Consequently, in trials, the system has achieved accuracies of 87.8% in obstacle detection and 85% in scene recognition [49].

DeepNAVI shows high portability and convenience because of its small model size, rapid inference time, and audio information delivery. Its onboard-trained deep learning models enable DeepNAVI to process information without an Internet connection. DeepNAVI differs from other devices in testing datasets. Similar devices [57,67] are developed based on existing general datasets or pre-trained object-detection models. DeepNAVI trains models using custom datasets, i.e., only obstacles and scenes relevant to the navigation domain. The current navigation system can detect twenty different types of obstacles and twenty indoor or outdoor scenes [49].

The low accuracy in object detection may be due to duplicate detection, misclassification, and/or mislocalization. There is still a need for DeepNAVI to improve its accuracy in detecting obstacles such as cabinetry, stairs, and white-colored objects. In addition, the scene recognition model should be enhanced for classrooms, construction sites, and building reception areas. The few instances of misclassification in scene recognition may be due to issues such as labeling ambiguities. A dataset with a wider variety of scenes and accurate labeling could mitigate these issues. Further work is needed on the distance estimation module so as to address the high deviation encountered with smaller obstacles (approximately 50 cm height by 50 cm width), such as chairs, tables, and waste containers. Moreover, the distance estimation model cannot detect obstacles more than five m away from users. When evaluating the motion detection and position estimation modules, a few misinterpretations are found in some moving objects. For further work, [49] suggest incorporating reinforcement components into deep learning models. Retraining these models with larger datasets can enhance the classification and detection accuracy of the navigation system.

DeepNAVI has been tested using twenty scenarios, and user surveys have been conducted. Numerous improvements for developing AI-based assistant systems, such as DeepNAVI, have been recommended [49]:(i)the system should detect more obstacles;(ii)adjustable feedback settings such as rate, pitch, or voice;(iii)multiple feedback modes such as haptic in addition to audio;(iv)avoid interruptions by another smartphone application while navigating.

Subsequently, thirteen participants were invited to evaluate the performance of DeepNAVI and provide qualitative feedback [59]. This required them to carry the DeepNAVI systems and then complete navigation tasks both with and without white canes. There were two navigation tasks in different natural environments: indoor environment (various physical obstacles such as chairs) and outdoor environment (multiple obstacles such as traffic signs, plants, and people). At the end of each task, the participants completed a semi-structured interview. Their responses were qualitatively coded by a thematic coding strategy (including inductive and deductive codes) [59].

In previous need-finding studies, one of the key expectations from VIPs is a portable navigation assistant [72,73]. During the DeepNAVI tests, participants provide favorable feedback because of its high portability (convenience, low weight, small form factor). However, participants require more information to assist navigation, such as the type and distance of obstacles. A few participants commented on both false obstacle detection and their reduced confidence in DeepNAVI in crowded public places. Moreover, they suggested additional modes of information delivery, such as beep tones and vibrations.

In summary, several future research directions exist for AI-based navigation assistants, such as DeepNAVI [59]:(i)an information moderation filter can provide cognitive power to process a considerable amount of information and further refine the navigation systems;(ii)provision of customization options for users to obtain the information they need during navigation;(iii)privacy protection: Ref. [74] has emphasized the challenges and requirements of VIPs in understanding the information relative to privacy under different techniques;(iv)ability to navigate in areas where data traffic and network coverage are low;(v)a trade-off between high accuracy and low latency in some lightweight deep-learning models [75].

In conclusion, technologies such as artificial intelligence offer enhanced convenience and performance for navigation systems. However, privacy concerns are increasingly important for VIPs when selecting their preferred navigation assistance. Various methods currently exist to ensure data privacy in navigation systems, e.g., external networks or cloud-based services can process information about the surrounding environment during navigation [58]. Stringent privacy standards are necessary to prevent potential privacy leakage when uploading data to external servers. Another effective method for safeguarding data privacy is processing and storing data locally on devices [49].

### 6.4. Summary of Navigation Systems for Visually Impaired People

The review above has highlighted a number of drawbacks in the current navigation systems for VIPs.

First, most of the previous research lacks a comprehensive definition of the positioning/navigation requirements for VIP applications. In particular, none of the research has outlined the unique needs and challenges of VIPs. A comprehensive investigation of VIPs’ positioning/navigation requirements is necessary for developing an effective system. Future research should examine how proposed systems address the full spectrum of visual impairments, focusing on inclusive designs for different VIP subgroups. Additionally, it is necessary to monitor the latest advancements in artificial intelligence and their impact on the development of navigation assistant systems for VIPs.

Second, the positioning accuracy and integrity of the current VIP navigation systems are insufficient to achieve safe navigation in urban areas. Table 9 and Table 10 show that the highest positioning accuracy is 20 cm from the infrared-based system [47] and the UWB-based system SUGAR [41]. Systems with this level of accuracy can struggle to meet stringent positioning requirements for detecting objects such as keys, leaving VIPs in hazardous navigation environments. Infrared sensors have limitations such as high sensitivity to sunlight, and UWB systems are expensive to construct in small-scale environments [41]. To address this, further research needs to explore advanced sensor fusion algorithms, which combine data from various sources, such as GNSS, inertial, visual (SLAM), etc.

Third, there is a lack of investigation in the transition areas in recent research. These transition areas, such as the entrance of a building, pose considerable challenges for VIPs’ navigation. The danger of navigation in transition areas comes from factors such as temporary signal interruptions, as well as multipath interference. To bridge this gap and achieve a seamless navigation aid for VIPs, research can concentrate on integrating different sensors, such as visual, inertial, indoor, etc.

Finally, almost all the experiments of the systems listed in Table 9 and Table 10 occur in quiet, indoor environments or with predetermined obstacles in the pathways. None of the VIP navigation systems considered more complex environments such as underground stations or more adverse weather such as storms. Detailed tests of the systems with different sensors under different environments remain absent, thus presenting an important research gap in the existing research.

## 7. Adaptation of Navigation Systems in Response to a Global Pandemic

The CORONAVIRUS disease 2019 (COVID-19) pandemic posed a significant health challenge globally, resulting in widespread lockdowns and other restrictive measures. Numerous surveys have been conducted across different countries to assess the impact of COVID-19 on various aspects of the lives of Visually Impaired People (VIPs). This section first examines the effects of the pandemic on VIPs and subsequently discusses how technology can assist them in navigating during such challenging times. Finally, it highlights the importance of developing advanced technology and simulation models to better prepare for future pandemics.

### 7.1. Impact of COVID-19 on Visually Impaired People

During the COVID-19 pandemic, social distancing and transportation difficulties were major concerns, particularly for those with greater visual impairments (VI). Comprehensive reviews, such as [76], highlight the difficulties of social distancing for VIPs, including:(i)visual limitations hinder their ability to maintain safe distances [77];(ii)they cannot detect directional arrows on floors;(iii)guide dogs cannot assess new distances;(iv)canes are only forward-facing probes.

Consequently, individuals with VI often approached closer to other individuals than sighted people, particularly in busy environments such as streets, subway platforms, and queues [78]. Studies have shown that VIPs create mental maps to navigate environments [79]. However, new layouts in public spaces during the pandemic can disrupt these mental maps, making navigation more challenging.

Surveys by [80] aim to investigate challenges that most visually impaired people face due to the COVID-19 shutdown. The questionnaires were completed by 232 participants. The results indicate that the severity of VI greatly influenced the types and magnitudes of challenges faced. Blind participants were more likely to view their vision as a risk factor for contracting COVID-19 and experienced worse outcomes. Those with moderate to severe VI and blindness reported greater difficulties in accessing eye care, transportation, and financial stability. These findings underscore the challenges individuals with substantial VI faced in adhering to the strict social distancing and hygiene protocols to prevent COVID-19 transmission.

Online surveys by [81] studied the effects of lockdown on Hungarian adults with visual impairments (N = 132). They examined access to shopping, daily support needs, remote education or work, and leisure habits. The results indicate that social distancing leads to a reduction in support requiring physical contact. Text responses illustrate that shopping for essential goods is the area where the most sighted support, without physical contact, was needed. Additionally, some responses show a lack of sufficient or any assistance.

### 7.2. Technology to Help Visually Impaired People Navigate

The pandemic underscored challenges in accessibility and independence for VIPs during navigation [82]. Technology has provided solutions, such as ‘Active Crowd Analysis’, which is a smartphone-based system that detects crowd risks through crowd density and motion analysis. The system then informs users about crowd risks via directional 2D audio to maintain social distancing [83]. Modifications of previously discussed vision-based systems can aid VIPs during pandemics.

Another app, ‘Be My Eyes’, offers remote-sighted assistance to VIPs during pandemics. It carries out live video-streaming and connects VIPs with sighted volunteers to describe their surroundings via phone cameras [84]. Practical challenges to its widespread use include limited training of volunteers and the unpredictable risk of losing Internet connectivity.

These technologies leverage existing hardware to enhance VIPs’ safety and independence during pandemics. To develop viable solutions, it is crucial to survey current assistive technologies and viable options to ensure VIPs’ safety and agency [78]. Future urban planning should integrate lessons from COVID-19 using simulation modeling to better prepare for similar challenges [85].

## 8. Conclusions and Future Research

All too often, the limited mobility of VIPs greatly affects their quality of life. Moreover, while ISO standards exist to provide them with greater opportunities for increasing their mobility, these suffer from drawbacks that stifle their potential, such as a lack of integration between appropriate software and physical navigation aids. In order for VIPs to fully participate in all activities of society, a high quality, reliable positioning system for their navigation and wayfinding is a pre-requisite. However, a problem arises in that findings in the medical and engineering fields relating to the mobility of VIPs are segregated. This is compounded by the wide spectrum of visual impairment which must be taken into account when devising any navigation and wayfinding system.

This paper attempts to overcome this challenge by initially examining the fundamental mobility requirements for VIPs and, based upon a thorough literature review, derives sixteen major requirements. Unfortunately, behavioral insights to inform VIPs’ mobility needs are often lacking. This paper, therefore, outlines the results of a series of experiments in a specially designed environment at PAMELA. This involved VIPs in various operating scenarios, providing crucial behavioral information for the design of any positioning system for VIP navigation and wayfinding. Based on the fundamental mobility requirements and the VIP behaviors examined, this paper assesses current positioning systems, highlights four major limitations, and provides avenues for further research.

Such research can enhance high-precision positioning technology for VIPs by conducting behavioral experiments in diverse environments. Table A1 in Appendix A provides a summary of the main methodology risks and their mitigation strategies. To address these risks, particularly the low positioning accuracy in GNSS-denied environments, potential improvements include:Quantifying the navigation system requirements, given that current knowledge has limitations in defining these for VIPs. This includes accuracy, integrity risk, alarm limits, availability, and continuity. These factors can be derived from user requirements, VIP behaviors, infrastructure environments, and risk analysis. This process may lead to various requirement categories based on the visual function of the VIP and/or their operating environments;Quantifying the situational awareness requirements to ensure safety, including the integrity budget;Implementing advanced carrier phase GNSS positioning technologies, such as RTK (Real-Time Kinematic) and Precise Point Positioning (PPP), as the current understanding of requirements indicates that a centimeter to decimeter accuracy level is necessary for VIPs’ navigation;Developing an integrity monitoring algorithm that supports carrier phase positioning, namely Carrier Phase Receiver Autonomous Integrity Monitoring (CRAIM);Developing an integrity monitoring layer for indoor positioning systems. In addition to Fault Detection and Exclusion (FDE) and computing the protection level, this improvement involves identifying faulty modes and models and overbounding the error distribution to ensure safety;Conducting a feasibility assessment for indoor positioning systems that can meet VIPs’ navigation requirements. This assessment should follow the system requirements and use the protection level value of each indoor positioning system;Developing an integrity monitoring algorithm for SLAM methods in case the feasibility assessment suggests using SLAM;Performing a cost–benefit analysis for the candidate indoor positioning systems;Developing a situational awareness layer that ensures safety.

Alongside technical advancements, it is crucial to prioritize user comfort in the development of assistive technologies. Potential directions include cost reduction (cost-effectiveness), reduced size and weight to enhance scalability, shortened learning times, implementation of a multimode feedback system, real-time information delivery, and privacy protection. Future research should include interviews or surveys to better understand the needs and preferences of VIPs, as suggested by [61]. These research efforts can make smart cities more inclusive for visually impaired communities. By applying the findings across disciplines, we can improve the accessibility of technologies such as autonomous vehicles for visually impaired groups.

## Figures and Tables

**Figure 1 sensors-24-07020-f001:**
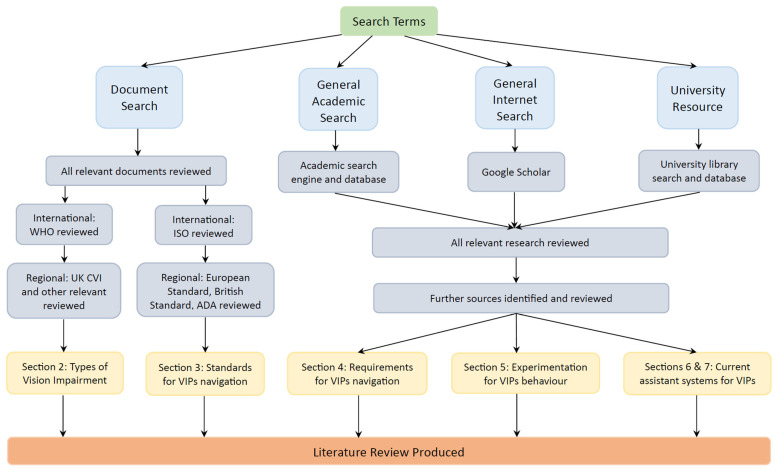
Methodological framework for the comprehensive literature review.

**Figure 2 sensors-24-07020-f002:**
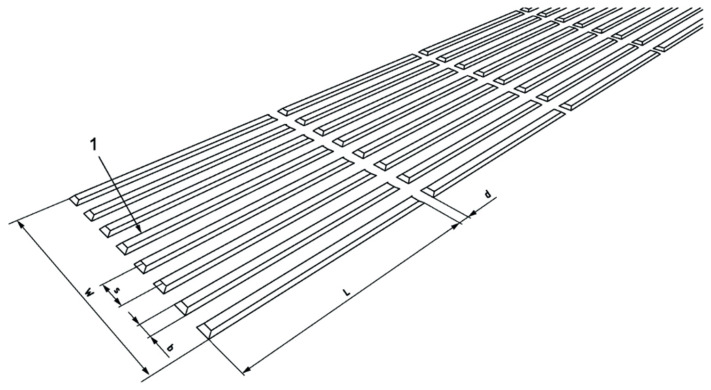
Standard of dimensions of TWSI [17]. Key 1: flat-topped elongated bars, height 4 mm to 5 mm, beveled; s: spacing of ribs; b: width at base; L: minimum 270 mm; W: minimum 250 mm; d: minimum 30 mm.

**Figure 3 sensors-24-07020-f003:**
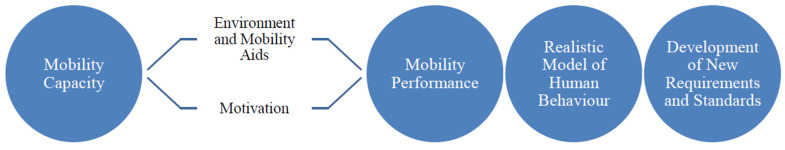
Transitioning from mobility capacity to the development of new requirements [5].

**Figure 4 sensors-24-07020-f004:**
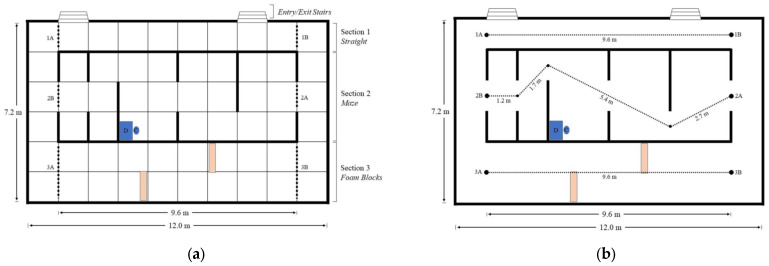
Mobility course configuration and walking path on the platform at PAMELA [5]: (**a**) sections of the PAMELA platform; (**b**) proposed walking paths of VIPs.

**Figure 5 sensors-24-07020-f005:**
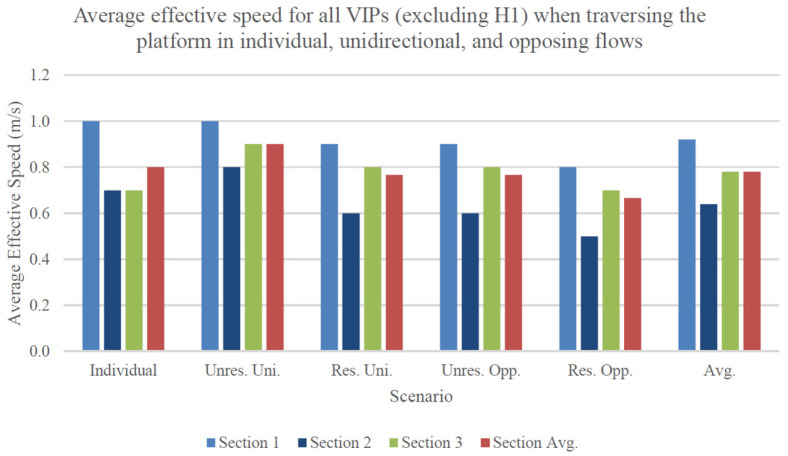
Average (avg.) effective speed for all VIPs when traversing the platform in individual, unidirectional, and opposing flows [5].

**Figure 6 sensors-24-07020-f006:**
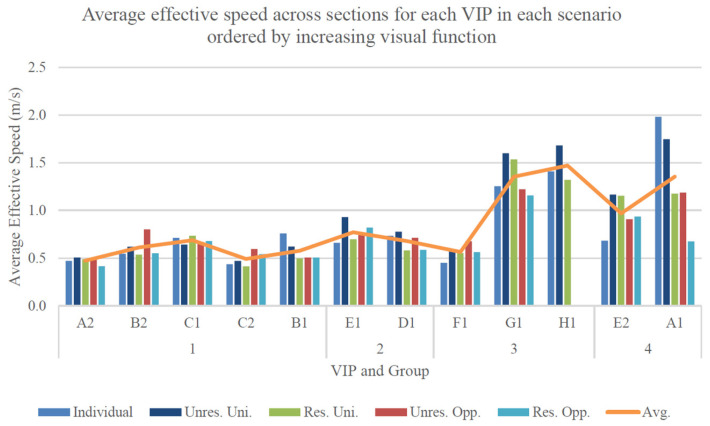
Average effective speed across sections for each VIP in each scenario ordered by increasing visual function [5].

**Figure 7 sensors-24-07020-f007:**
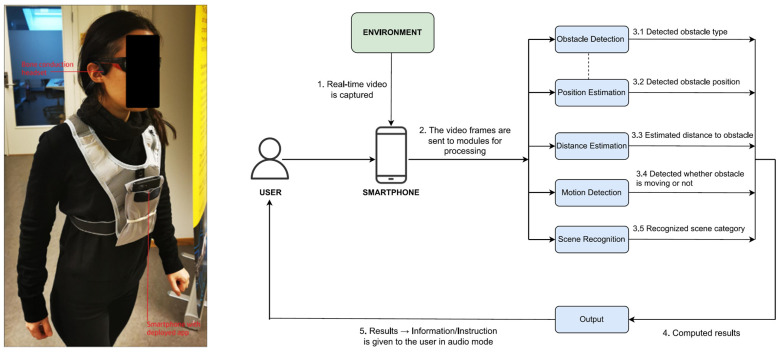
Components of the DeepNAVI navigation assistant system [49].

**Table 1 sensors-24-07020-t001:** ICD classification of visual impairment severity [2].

Classification of Visual Impairment	Visual Acuity Worse Than
Distance—Mild	6/12
Distance—Moderate	6/18
Distance—Severe	6/60
Distance—Blindness	3/60
Near Vision Impairment	N6 or M.08 with existing correction

**Table 2 sensors-24-07020-t002:** Descriptions of test samples in a selection of studies investigating visually impaired mobility [5].

Reference	Title	Location	Test Sample
[7]	The Characteristics of Blind and Visually Impaired People Evacuation in Case of Fire	Russia	201 visually impaired participants (68.2% over 40 years of age) classified according to Russian medical legislation where all people with disabilities are divided into 3 groups depending on their functional system of the body with the most severe problems relating to group I and the least to group III.
[8]	Evacuation characteristics of visually impaired people—a qualitative and quantitative study	Denmark	40 visually impaired participants aged 10–69 years were classified according to Danish classification (based on visual acuity ‘x’) as follows: A—Visually Impaired (0.1 < x ≤ 0.3) B—Social blindness (0.01 < x ≤ 0.1) C—Practical blindness (0.001 < x ≤ 0.01) D—Total blindness (x ≤ 0.001)
[9]	Mobility-related accidents experienced by people with visual impairment	United States	307 visually impaired survey participants aged 18 years and older were asked to describe their level of vision loss as either ‘blind’ (at most some light perception) or ‘legally blind’ (legally blind but not blind).
[10]	Mobility in Individuals with Moderate Visual Impairment	United States	22 ‘low vision’ participants aged 19–58 were tested on visual acuity, visual field, and contrast sensitivity.
[11]	Does Visual Impairment Affect Mobility Over Time? The Salisbury Eye Evaluation Study	United States	2520 Salisbury Eye Evaluation Study participants aged 65 years and older followed over 2, 6, and 8 years after baseline. Visual impairment was defined as best-corrected visual acuity worse than 20/40 or a visual field of approximately less than 20°.
[12]	Emergency lighting and wayfinding provision systems for visually impaired people: Phase I of a study	United Kingdom	30 participants aged 36–73 years were classified only according to their cause of visual impairment, including RP, Macular degeneration, Glaucoma, and Diabetic retinopathy.
[13]	Association of Visual Impairment with Mobility and Physical Function	United States	Interview of 5143 older residents aged 65 years and older from three communities (Established Populations for the Epidemiologic Studies of the Elderly), classified according to visual acuity screening, self-reported activities of daily living and mobility, and objective physical performance measures of balance, walking, and rising from a chair.
[14]	Effect of obstacle density on the travel time of visually impaired people	China	8 sighted and 32 visually impaired participants were classified as ‘near blindness’ by Chinese national standards of 28–55 years. The range of vision loss (based on visual acuity) of the visually impaired people was <0.05, and they were described as being unable to walk without a walking stick or assistance from family members.
[15]	Visual function, visual attention, and mobility performance in low vision	United States	35 visually impaired participants aged 20–80 years with low vision due to various visual disorders classified according to UFV and clinical measures of contrast sensitivity, visual field, and visual acuity. Two models were considered; series 1 used the UFV scores as measured, and series 2 used the UFV scores corrected for visual field loss (only counting errors in areas of intact visual field).
[16]	Limitations in mobility: experiences of visually impaired older people	The Netherlands	10 visually impaired older participants aged 63–95 at a Dutch center for visual impairment with the following inclusion criteria: 1. Had applied for mobility training at the center 2. Had visual deficiencies 3. Were experiencing problems in mobility as a pedestrian 4. Had never before received mobility treatment 5. Had received treatment from someone other than the first researcher 6. Did not have any of the following problems: deafness or secondary psychiatric disabilities, or severe physical, diabetic or neurological disabilities 7. Spoke and understood Dutch 8. Were willing to participate in the study

**Table 3 sensors-24-07020-t003:** Minimum luminance contrast values for different visual tasks [17].

Visual Task	$\mathrm{Minimum}\;{LRV}^{\left(1\right)}$ of the Lighter Surface (CIE Y) [%]	$\mathrm{Michelson}\;\mathrm{Contrast}\;C_{M}$ [%]	$\mathrm{Weber}\;\mathrm{Contrast}\;C_{W}$ [%]
Large surface areas (i.e., walls, floors, doors, ceiling), elements and components to facilitate orientation (i.e., handrails, door furniture, tactile walking surface indicators, and visual indicators on glazed areas)	≥40	≥30	≥45
Potential hazards (i.e., visual indicators on steps, glazed areas), small items (i.e., switches and controls), and self-contrasting markings	≥50	≥60	≥75
Text information, i.e., signage	≥70	≥60	≥75

^(1)^ The Light Reflectance Value (LRV), or CIE Y-value, is expressed on a scale of 0 to 100, with a value of 0 points for pure black and 100 points for pure white.

**Table 4 sensors-24-07020-t004:** Minimum light levels in different areas [17].

Different Areas	$E_{min}$ [lx]
Outdoor environments	20
Horizontal surfaces indoors	100
Stairs, ramps, escalators, moving walkways	150–200
Habitable spaces	300–500
Visual tasks with small details or low contrast	1000

**Table 5 sensors-24-07020-t005:** Summary of standards for VIPs navigation.

Standard Title	Key Details for VIPs	Limitations	Reference
ISO 21542: Building construction. Accessibility and usability of the built environment.	Standards in tactile, visual, and audible information for VIPs orientation, emergency warning system, minimization of obstacles and hazards in the pathway, standards of relief facilities for assistance dogs	1. Lack of standards for elements of external environments 2. Lack of discussion of regional variations in VIP requirements and local regulations 3. Complex and costly to implement all standards 4. It does not account for the rapid emergence of navigation aids for VIPs 5. Insufficient incorporation of VIP feedback on real navigation needs	[17]
ISO 9241-171: Ergonomics of human-system interaction	Guidelines on names and labels for user-interface elements, user preference settings, user preference settings, special considerations for accessibility adjustments, controls and operations, compatibility with assistive techniques, etc.	1. Lack of detailed discussion on real-world requirements of VIPs 2. Lack of guidance on the integration of physical aids such as tactile maps	[18]
European Standard EN 301 549	Comprehensive guidance on improving the accessibility of ICT products and services, alignment with WCAG	1. Neglect considerations for physical aids 2. Struggle to match the pace of evolving technologies for accessibility	[19]
2010 ADA Standards for Accessible Design	Outlines both scoping and technical requirements for accessibility; crucial standards such as the removal of barriers at lift call buttons, speech-output-enabled machines, tactile signage, and accessible routes	1. Irregular revision frequency 2. A lack of detailed exploration of how the navigation needs to change with different levels of vision impairment 3. Impracticality in international regions	[20]
2018 BS-8300: Design of an accessible and inclusive built environment	Provides comprehensive guidelines for designing accessible environments for VIPs in the UK, BS 8300-1 for external environments, BS 8300-2 for buildings	1. Significant implementation costs 2. Efforts of regular training and maintenance	[21,22]

**Table 6 sensors-24-07020-t006:** Requirements for generic environmental features [5].

Feature	Literature Gap	Hard User Requirement	Soft User Requirement
Entrance/Exit	- Entrance/exit type (such as a revolving or push/pull door)	- Accessibility of entrance/exit according to user ability - Usability as an entrance, exit, or both	- Prerequisites to accessing these (such as a key card)
Pathway	- Walking position (left/right) in one-way systems - Texture and gradient	- Directionality and walking position	- Texture information for initial user planning
Crossing	- Length of crossing - Green time	- Controlled, uncontrolled, or spontaneous crossings	- Length of crossing - Green time
Decision Point	- Accessibility according to user ability - Emergency egress scenarios	- Accessibility of decision points according to user ability	- Egress routes and nearest exit upon alarm
Tactile Surfaces	- Variability and inconsistency of existing tactile surface standards and applications	- Do not identify tactile surfaces to the user	- Surface textures, such as a smooth surface, to indicate entry points and routes
Escalator	- Escalators may change movement direction (up/down/static) - Up/down/static escalators may vary in position	- Escalator movement direction - Escalator position	- Escalator width as a user preference - Handrail position - Distance to landing
Stairs	- Stairs incorporate different tactile surfaces and edges - Navigating stairs is a repeated action forming tacit knowledge	- Do not identify the number of stairs or tactile surfaces	- Availability of handrail and its position - Head height obstacles or landing height - Alert for open riser staircase
Lift Lobby	- New lifts with a centralized lift call	- Number of lifts - Route guidance to the arriving lift	- Connections between lift systems (call lift)
Required Lift	- Not considered in [26] standard	- Positioning and ordering of buttons	- Presence of an accessibility button
Toilet	- Not considered in [26] standard	- Toilet type and accessibility (such as a changing places toilet)	- Locks or interactivity (such as a radar key or code)
Queuing Point	- Not considered in [26] standard	- Queuing point structure (structured/unstructured)	- Typical queue length
Gateline	- Span of gateline and position of entry/exit gates	- Span of gateline and position of entry/exit gates	- Points at which not to stand or wait
Platform	- Platform width - Seating location - Shelter availability - Help point location	- Platform width and edge warning - Seating location - Help point location	- Shelter availability - Customer information display location - Availability of platform edge doors - Illumination level

**Table 7 sensors-24-07020-t007:** Self-reported visual function of the VIPs (CVI: Certificate of Visual Impairment, PVL: Peripheral Vision Loss, SI: Sight Impaired, SSI: Severely Sight Impaired, VA: Visual Acuity, CS: Contrast Sensitivity, VF: Visual Field, SD: Standard Deviation, CV: Coefficient of Variation) [5].

ID	CVI	PVL	Impairment	VA	CS	VF	Avg.
A1	SI	Y	Cataracts, Nystagmus, Detached Retina, Glaucoma	2	2	1	1.7
A2	SSI	Y	Still’s Disease	1	1	1	1.0
B1	SSI	Y	Retinopathy of Prematurity—Retinal Fibroplasia	1	1	1	1.0
B2	SSI	Y	Optic Nerve Atrophy	1	1	1	1.0
C1	SSI	Y	Retinitis Pigmentosa	1	1	1	1.0
C2	SSI	Y	Retinitis Pigmentosa	1	1	1	1.0
D1	SSI	N	Congenital Heredity Endothelial Dystrophy	1	1	1	1.0
E1	SSI	Y	Retinitis Pigmentosa	1	1	1	1.0
E2	SI	N	Glaucoma, Detached Retina	1	2	1	1.3
F1	SSI	Y	Retinitis Pigmentosa	1	1	1	1.0
G1	SSI	Y	Glaucoma	1	1	2	1.3
H1	SSI	N	Stargardt Disease	1	1	2	1.3
Avg.				1.1	1.2	1.2	1.1
SD				0.3	0.4	0.4	0.2
CV				26.6	33.4	33.4	19.6

**Table 8 sensors-24-07020-t008:** Spread of average (avg.) effective speed for all VIPs (H1 excluded) when traversing the platform in individual, unidirectional (uni.), and opposing (opp.) flows (SD: Standard Deviation, CV: Coefficient of Variation, Res.: Restricted, Unres.: Unrestricted) [5].

Metric	Individual	Unres. Uni.	Res. Uni.	Unres. Opp.	Res. Opp.
Avg. (m/s)	0.8	0.9	0.8	0.8	0.7
SD (m/s)	0.2	0.1	0.2	0.2	0.2
CV (%)	21.7	11.1	19.9	19.9	22.9

**Table 9 sensors-24-07020-t009:** Current navigation systems for VIPs (with Vision Systems).

System Category	System Name	Sensors/Main Components	Performance: Range/Accuracy/Other	Advantage(s)	Limitation(s)	Reference
RGB-D camera (vision)	Not Applicable	Glasses with Realsense D435i camera embedded computer, headphones, microphone	Target detection success rate of 95.46%, optimal paths prediction success rate of 92.60%	1. Novel fusion neural network aids VIPs in spatial target localization 2. The path planning method combines neural networks to find the optimal route based on fusion data 3. Wearable navigation system showcases visual-auditory fusion and navigation for VIPs	1. Enhancement of battery life and exploration of alternative power sources 2. Emphasis on design improvements for E-glasses, prioritizing comfort and aesthetics. 3. Limited investigation into the real-world requirements of VIPs	[34]
RGB-D camera (vision)	ISANA	Embedded RGB-D camera, wide-angle camera (visual motion tracking), 9-axis IMU	Accuracy: ISANA localization error smaller than 1 m without drifts (Closed loop)	1. Indoor wayfinding for VIPs with environmental awareness, obstacle detection, and avoidance, robust HMI 2. Average navigation guidance error decreased significantly by adding SmartCane for accurate heading measure, and traveling time decreased for the same travel	1. User interface to be improved, such as better speech recognition to achieve robustness in noisy places; audio frequency should be adjustable and customized 2. Lack of ability to navigate in complex, cluttered environments such as underground stations	[35]
Raspberry Pi camera	Not Applicable	Raspberry pi 4 (4 GB RAM), ultrasonic sensor (HC-SR04), Raspberry Pi camera (V2), servo motor (sg90), battery	High object detection probability (e.g., 0.97 for a suitcase), average efficiency of 91%	1. Efficient in terms of battery life, night vision, and weight 2. Simple and easy-to-use algorithms do not require internet access to operate 3. Clear image captured with minimum noise 4. Detection of the clear demarcation between the hindered object and the surroundings for VIPs to walk easily 5. Successful detection of the periphery of objects (e.g., transparent bottles)	1. Possible faulty object detection because the picture is outside the frame 2. Distortion in the captured image when one VIP is moving 3. Hard to differentiate transparent objects and the background	[36]
Stereo camera (vision)	Not Applicable	Binocular camera (environment perception), visual odometry (head tracking), IMU	Sensing range: 10–20 m; Positioning accuracy	1. Environmental perception (through modeling geometric scene background by the binocular camera) 2. A robust head tracking algorithm (combines IMU and visual odometry) is used to obtain highly frequent and robust head pose estimations (head orientation and position) with small latency 3. Sonification of objects in users’ vicinity to overcome limitations of other assistive aids, semantically meaningful (different sounds to distinguish object categories), and comfortability 4. Enable users to walk in (unknown) urban areas and prevent obstacles 5. Wearable, lightweight, unobtrusive	1. Although sonification can notify users about dangerous objects, the lack of decision of direction to walk will increase the risks of danger 2. The bulky helmet with camera and backpack must be minimized 3. Lack of significant evidence for operations indoors 4. Sonification ignores the needs of deafblind people	[37]
Visual SLAM	Virtual-Blind-Road Following-Based Wearable Navigation Device	Depth camera, fisheye camera, ultrasonic rangefinder	Accuracy: Average deviation distance less than 0.3 m with a small variance	1. Route following (PoI graph + dynamic subgoal selecting-based route following algorithm) and obstacle avoiding problems solved simultaneously 2. Low-cost, small dimension, relatively easy integration	1. Higher accuracy is required in other applications than walking 2. The continuity of the system for replying only on cameras (e.g., worse performance at nighttime)	[38]
LiDAR (vision)	LASS	LiDAR device, stereo headphone set (Audio Technica AUD ATHAD500X), scanning laser rangefinder (Hokuyo URG-04LX)	LASS system performs better (higher obstacle hit rate) in closer obstacles and when obstacles are found at a 90-degree angle	1. Compared with ultrasound as the signal source, LiDAR is better due to its shorter wavelength and focused beam, thus higher spatial resolution 2. LiDAR can constantly scan the surrounding environments without head movements 3. LASS translates the distance information of the object to sounds with different pitches, which can be optimized to users’ reactions (closer objects with higher frequencies) 4. Wearable LiDAR device 5. Less training time 5. LASS can generate and receive signals so users can focus on interoperating spatial information	1. Difficult to use the laser pointer to point at targets; longer training time is needed for the users 2. Long time cost for full scans 3. The complex layout of obstacles in the real world	[39]

**Table 10 sensors-24-07020-t010:** Current navigation systems for VIPs (without Vision Systems).

System Category	System Name	Sensors/Main Components	Performance: Range/Accuracy/Other	Advantage(s)	Limitation(s)	Reference
Bluetooth	Hybrid Indoor Positioning System for VIPs Using Bluetooth and Google Tango	BLE system (Samsung Galaxy S4), Tango system (Lenovo Phab 2 Pro)	(Hybrid system): Hybrid system requires less aid from researchers than BLE-only navigation system (*p* = 0.0096) but similar travel duration	1. Able to create detailed 3D models of the indoor environment by utilizing the Tango system (which has an RGB-D camera for feature-based indoor localization and pose estimation) 2. Sensors of Tango are sensitive to noise 3. Tango can try to correct the accumulated drift errors 4. Vibrotactile aid (different vibration speed depends on how close the object is)	1. BLE signals are extremely noisy due to the attenuation by materials inside the building. Fingerprinting method to compare current Received Signal Strength Indicator (RSSI) and pre-built snapshots of the area’s radio landscape, but the beacons can only produce a coarse location for users, so greater accuracy is required for the localization 2. The Tango system can only provide feature maps for around one floor each time (BLE can help to determine which ADF Tango to load to report accurate position to users) 3. Future works in supplements such as vibrotactile	[40]
UWB	SUGAR	wall-mounted Ubisense UWB-based sensors, smartphone, UWB tag embedded in headphones, server with various modules	Accuracy: positioning errors up to 20 cm in most cases	1. UWB can distinguish direct-path signals from reflected signals 2. UWB combines the position estimation method to diminish the effects of obstacles 3. High accuracy for obtaining the user’s location 4. Voice commands and acoustic signals to guide users to the selected destinations	High expense in small-scale deployments, the cost can limit its usage for private homes	[41]
RFID	Mobile Blind Navigation System Using RFID	Android-operated mobile, RFID reader on white cane, GPS, RFID tags	High performance in obstacle avoidance and blind guidance	1. Lightweight of the RFID handheld reader 2. Voice recognition and text-to-speech to communicate with VIPs 3. High performance in obstacle avoidance	1. Cost of deployment of the RFID tags 2. Lack of detection of moving obstacles 3. Lack of experimentation to test the efficiency of the system in different environments	[42]
Ultrasonic	Navguide	Six ultrasonic sensors (all are wide beam ping sensors for detecting floor-level and knee-level obstacles), one wet floor detector sensor, one step-down button, micro-controller circuits, four vibration motors, one battery	Under most experiment sets, Navguide reduces the rate of collision with obstacles; participants with Navguide can complete the tasks faster compared with participants using a white cane	1. Main goal is to supply logical maps of environments and feedback on obstacles to users 2. Ability to detect wet floors to prevent slipping accidents 3. Detections of floor-level and knee-level obstacles 4. Tactile and auditory sensing are available for simple and priority information 5. Cheap and light to carry	1. Unable to detect any pit and downhill 2. Unable to detect downstairs, higher risk of falling 3. Only sense the wet floor when users step on it	[43]
Wi-Fi	INAVI	Wi-Fi signal scanner (NodeMCU ESP8266), a speaker	Location recognition accuracy of 96%, an average localization accuracy of 1.5 m (with more checkpoints)	1. Leverage the Wi-Fi infrastructure within most public buildings to reduce cost 2. Ensure maximum portability with minimal hardware requirements (only ESP8266 and a speaker)	1. Long response times occur without Wi-Fi signals 2. Wi-Fi signal fluctuations affect RSSIs 3. Trilateration needs three signals, limiting coverage in smaller areas 4. Users lack obstacle guidance, posing challenges	[44]
Haptic	HaptiSole	Four vibrator motors (links to the board of Arduino) on the sole, Bluetooth transmission module, battery	An average accuracy rate of 94% in identifying four main directions (e.g., up, forward, right, left)	1. Users can accurately recognize directions displayed on their feet 2. Blindfolded users can reach destinations 3. The system is embedded in shoes, and directions are communicated by vibration motors 4. An evaluation platform that helps EOA designers evaluate user interfaces without positioning errors 5. Considerations for ergonomics, self-sufficiency, resolution, aesthetics	1. Users faced challenges in recognizing up-right and up-left directions 2. Multiple outages occurred during testing 3. The performance under diverse environmental and ground conditions remains unexplored	[45]
Smartphone-based	NavCog3	Smartphone (with in-built IMU, BLE beacons)	Accuracy: average accuracy is 1.65 m; performance: 93.8% of 260 turns are successful, more difficult to turn by 45 degrees than 90 degrees	1. Provide turning instructions to the users 2. Instant audio feedback when the orientation is incorrect 3. Supply information such as distance, location, and direction of landmarks and points of interest 4. Compared with LiDAR, ultrasonic, UWB, etc., NavCog3 does not need more signal receivers	1. For large-scale environments, there is a high deployment cost, and it is time-consuming to collect landmark information indoors 2. Lack of control users’ exposure to earlier versions of the system, which can affect the accuracy or subjective responses	[46]
Infrared	Low Energy Precise Navigation System for the Blind with Infrared Sensors	1. Device with two infrared sensors (M1 × 90,614 with 90 and 35 degrees FOV) (measures infrared radiation) on VIPs’ arms 2. Gyroscope, accelerometer, magnetometer sensors (MPU9250 module) (for compensating the deflection of the system)	Accuracy: 20 cm (average accuracy compared with tap measurements)	1. Take advantage of different temperatures of walls and floors (or objects made of varied materials) to continuously find the distance between VIPs and objects such as walls and buildings, larger temperature differences when far from the object 2. Low power consumption 3. No need to emit signals	1. The accuracy depends on the resolution of the infrared sensors or cameras 2. Accuracy should be improved for more complex environments and weather	[47]

## Data Availability

Not applicable.

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
