# Peer review of "Comprehensive Review: High-Performance Positioning Systems for Navigation and Wayfinding for Visually Impaired People"

_sensors, 2024, doi:10.3390/s24217020_

Round 1
Reviewer 1 Report
Comments and Suggestions for Authors
This document presents a comprehensive review of mobility visual aids for the visually impaired. It reviews a number of mainly European documents, standards and recommendations. It also presents experiments to study combinations of aids and their effect on the walking speed of a visually impaired person.
Pros :
- there is severel technologies compared
- paper is easy to read
- validation of some technology combinations was realized
- experiments used concurrent and opposed flows of persons
Cons :
- the section on methodology refers to a systematic analysis of the literature, whereas the paper presents a comprehensive review with possible biases
- it is not clear whether the revision concerns the UK, Europe or international markets. In the latter case, a major revision is needed to incorporate more standards and recommendations.
- experiments lacks a number of technology combination and VIP to be complete, but Limitations explain this.
Corrections :
- please provide how you did your comprehensive review by stating search keywords, inclusion and exclusion criteria
- clarify if your paper is aimed at a specific area or is worldwide. In the later case, you should reconsider UK specificities like CVI and BS standards
- please add more up to date references. The earliest reference was in 2022.
- tables provide several details but each line does not seem to be balanced with each other. Also, some comments are not supported appropriatly by a reference or an experiment. For example, why UWB is more expansive than others ?
English is good. Only suggested term at line 17 is to replace the word "of" by the word "for" : The primary reason of these challenges lies in the segregation of Visual Impairment
Reviewer 2 Report
Comments and Suggestions for Authors
1: The paper comprehensively reviews the current state of navigation systems for visually impaired persons (VIPs) but lacks a clear integration of direct user feedback. Future research should include a more substantial qualitative component, such as interviews or surveys, to capture the nuanced needs and preferences of VIPs.
2: While the review touches on various types of visual impairments, it does not delve into how the proposed systems cater to the full spectrum of these impairments. It would be beneficial to see a more inclusive design philosophy that addresses the specific needs of different subgroups within the VIP community.
3: The paper mentions the development of high-performance navigation aids but does not thoroughly address the scalability and cost-effectiveness of these solutions. It would be insightful to have a dedicated section discussing how these systems can be made affordable and accessible to a broader audience.
4: The review should consider the rapid pace of technological advancements, particularly in the field of AI and machine learning. It would be advantageous to explore how these emerging technologies could be leveraged to improve the accuracy and reliability of navigation systems for VIPs.
5: The paper could benefit from a deeper analysis of cultural and regional differences in the application of navigation systems for VIPs. Different societies and regions may have unique challenges and opportunities that could influence the design and implementation of these systems.
6: With the increasing reliance on digital systems, the issue of privacy and data security becomes paramount. The paper should address how the proposed navigation systems ensure the privacy and security of user data, especially in the context of location tracking and personal information.
7: The COVID-19 pandemic has significantly altered public spaces and behaviors. It would be interesting to see an analysis of how the pandemic has affected the mobility of VIPs and how the proposed navigation systems can adapt to such global crises.
8: The paper mentions various experimental setups but lacks extensive real-world testing. Future research should include validation studies in diverse real-world environments to ensure the practicality and effectiveness of the proposed systems.
9: The review highlights the segregation between medical and engineering disciplines. It would be beneficial to advocate for more interdisciplinary collaboration, bringing together experts from various fields to develop more holistic and effective navigation solutions for VIPs.
10: The paper should consider the importance of longitudinal studies to assess the long-term usability and effectiveness of navigation systems for VIPs. Understanding how these systems perform over time and adapt to changes in the users' conditions and environments is crucial for sustainable development.
Comments on the Quality of English Language1. Consistency in Tense Usage: The manuscript would benefit from a consistent application of verb tenses throughout the text. In particular, the abstract and introduction sections occasionally shift between past and present perfect tenses, which can be confusing for readers. Ensuring a uniform tense will improve the clarity and flow of the writing.
2. Refinement of Sentence Structure: Some sentences in the paper are quite lengthy and complex, making them difficult to follow. Consider breaking these down into shorter, more digestible sentences to enhance readability. Additionally, watch for instances where a sentence may start with an introductory phrase but then shifts awkwardly into the main clause.
3. Subject-Verb Agreement: There are a few instances where the subject-verb agreement is inconsistent. For example, "The team *has* conducted" should be "The team *conducted*" if referring to a completed action. Ensuring that the subject and verb agree in number and tense will help maintain grammatical accuracy.
4. Proper Use of Articles: The manuscript occasionally omits or misuses articles, which is crucial for the clarity of noun phrases. For example, "VIPs navigation systems" should be "VIPs' navigation systems" to indicate possession. Pay close attention to the use of 'a', 'an', and 'the' to ensure that every noun is properly modified.
5. Punctuation for Clarity: The use of punctuation, particularly commas and semicolons, can be improved to enhance the clarity of complex statements. Commas are sometimes missing after introductory elements or before conjunctions that connect independent clauses. Semicolons can be used effectively to link closely related ideas within the same sentence. A careful review of punctuation will improve the overall readability of the manuscript.
Reviewer 3 Report
Comments and Suggestions for Authors
This review provides a comprehensive survey of navigation systems used by visually impaired persons and the classification of visual impairments. In addition, it gives gaps in current research and directions for future work. This review effectively summarizes previous research. However, this review still suffers from some typographical problems and an insufficient number of references. Thus, a major revision is required, and the final decision cannot be made until the following problems are solved.
The following are the comments.
1、 The title of this review is too long. The authors should condense and optimize the title to more accurately reflect the focus of the study.
2、 As a review, the number of references in this paper is insufficient. The authors need to refer to more literature and analyze it in depth.
3、 This paper contains some tables that have inappropriate layout issues. For example, tables 5 and 9 exceed the margins. Therefore, the authors should optimize the table layout in this paper.
4、 In this paper, some of the sentences are difficult to understand. For example, on page 1, the second sentence of the introduction is too long and difficult to understand; the author should simplify the sentence.
5、 On page twenty of this paper. The authors' assessment of gaps and recommendations in Literature 33 is inconsistent with the formatting of the other corresponding content. The author should revise the formatting of this part.
Comments on the Quality of English LanguageThis review provides a comprehensive survey of navigation systems used by visually impaired persons and the classification of visual impairments. In addition, it gives gaps in current research and directions for future work. This review effectively summarizes previous research. However, this review still suffers from some typographical problems and an insufficient number of references. Thus, a major revision is required, and the final decision cannot be made until the following problems are solved.
The following are the comments.
1、 The title of this review is too long. The authors should condense and optimize the title to more accurately reflect the focus of the study.
2、 As a review, the number of references in this paper is insufficient. The authors need to refer to more literature and analyze it in depth.
3、 This paper contains some tables that have inappropriate layout issues. For example, tables 5 and 9 exceed the margins. Therefore, the authors should optimize the table layout in this paper.
4、 In this paper, some of the sentences are difficult to understand. For example, on page 1, the second sentence of the introduction is too long and difficult to understand; the author should simplify the sentence.
5、 On page twenty of this paper. The authors' assessment of gaps and recommendations in Literature 33 is inconsistent with the formatting of the other corresponding content. The author should revise the formatting of this part.
Round 2
Reviewer 2 Report
Comments and Suggestions for Authors
1. The abstract should succinctly summarize the paper's objectives, methods, and conclusions. Consider revising it for enhanced clarity and to immediately engage readers.
2. Elaborate on the research methodology, particularly the criteria for selecting the studies reviewed. This will strengthen the paper's methodological rigor.
3. Provide a more detailed analysis of the limitations of existing technologies discussed in the paper, including specific examples and potential workarounds.
4. If applicable, use graphs or tables to present statistical data from the PAMELA experiments and other studies for better readability and comprehension.
5. Include a section on ethical considerations, especially regarding the use of personal data in navigation systems for visually impaired persons.
6. Expand the literature review to include a more diverse range of studies, potentially from less-represented regions or with different methodologies.
7. Ensure that all specialized terms are clearly defined upon first use to aid understanding for readers from varied disciplinary backgrounds.
8. The conclusions should not only summarize the findings but also provide actionable insights or recommendations for future research and practical applications.
9. Highlight how the paper's findings can be applied across different disciplines for a more comprehensive impact.
10. Ensure that all references are current and that any recent, relevant studies are included to reflect the most up-to-date research in the field.
Comments on the Quality of English Language
1. Ensure that verb tenses are consistent throughout the manuscript. For example, in the abstract, the phrase "The state-of-the-art VIP navigation systems cannot achieve" might be better phrased in the present perfect tense to reflect ongoing challenges, such as "The state-of-the-art VIP navigation systems have not yet achieved..."
2. Check for subject-verb agreement errors. For instance, sentences like "The challenges for these systems lies in..." should use "lie" instead of "lies" to agree with the plural subject.
3. Enhance the clarity of pronouns by ensuring they clearly refer to their antecedents. For example, in the sentence "This paper conducts a comprehensive review," the pronoun "this" could potentially be ambiguous. It should be clarified to "This paper, authored by the team, conducts a comprehensive review..."
4. Use parallel structure in lists or comparisons for better readability. For example, a sentence like "The review identifies gaps, offering insights, and provides recommendations" should maintain parallelism, such as "The review identifies gaps, offers insights, and provides recommendations."
5.Correct the use of articles to ensure grammatical accuracy. For instance, in phrases like "Visual Impairment reduces quality of life," consider whether an article is needed for specificity or generality, such as "Visual impairment reduces the quality of life."
Author Response
Please see attached file below

Reviewer 3 Report
Comments and Suggestions for Authors
No more comment
Comments on the Quality of English LanguageNo more comment
Author Response
See file below